# Synergistic sequence contributions bias glycation outcomes

Joseph M. McEwen[1], Sasha Fraser[1], Alexxandra L. Sosa Guir[1], Jaydev Dave[1] & Rebecca A. Scheck [1✉]

The methylglyoxal-derived hydroimidazolone isomer, MGH-1, is an abundant advanced glycation end-product (AGE) associated with disease and age-related disorders. As AGE formation occurs spontaneously and without an enzyme, it remains unknown why certain sites on distinct proteins become modified with specific AGEs. Here, we use a combinatorial peptide library to determine the chemical features that favor MGH-1. When properly positioned, tyrosine is found to play an active mechanistic role that facilitates MGH-1 formation. This work offers mechanistic insight connecting multiple AGEs, including MGH-1 and carboxyethylarginine (CEA), and reconciles the role of negative charge in influencing glycation outcomes. Further, this study provides clear evidence that glycation outcomes can be influenced through long- or medium-range cooperative interactions. This work demonstrates that these chemical features also predictably template selective glycation on full-length protein targets expressed in mammalian cells. This information is vital for developing methods that control glycation in living cells and will enable the study of glycation as a functional post-translational modification.

[1] Department of Chemistry, Tufts University, Medford, MA, USA. ✉email: rebecca.scheck@tufts.edu

Spontaneous, nonenzymatic modifications of proteins are hallmarks of cellular stress, disease, and aging[1–4]. In particular, the spontaneous attachment of sugars and sugar-derived metabolites to amino or guanidino groups on proteins occurs through a process known as glycation[5,6]. Glycation proceeds through a series of loosely understood condensation and rearrangement steps, resulting in a diverse set of modifications known as advanced glycation end-products (AGEs)[3,7–13]. To date, more than 25 distinct AGEs have been identified, which stem from the multitude of biologically relevant aldehydes[4,14,15]. The metabolic byproduct methylglyoxal (MGO), a 1,2-dicarbonyl, is one of the most reactive and abundant cellular glycating agents, with typical concentrations ranging from 1–5 μM and sometimes rising over 300 μM[16–18]. MGO is known to react preferentially with arginine (Arg) residues to produce numerous AGEs, including the methylglyoxal-derived hydroimidazolone isomers (MGH-1,-2,-3), carboxyethylarginine (CEA), argpyrimidine (APY), and tetrahydropyrimidine (THP) (Fig. 1)[19].

Glycation is a selective reaction that occurs at some, but not all, Arg or Lys residues and typically just a subset of AGEs form at each site (Fig. 1). Although past work has cataloged numerous instances of preferential glycation[20–28], there has been little investigation of the mechanism through which specific AGEs form on proteins[25]. Additionally, it remains an open question as to what controls the overall propensity for certain proteins to become glycated at certain sites[29,30]. Recent work from our lab has demonstrated that protein sequence is a primary determinant that governs the likelihood that a site will become glycated, and has also revealed that both sequence and folded structure sculpt the distinct complement of AGEs that form[29].

The MGH isomers are among the most commonly observed AGEs and are implicated in many diseases including diabetes, cardiovascular disease, and age-related disorders of the skin and eye[8,31–33]. MGH modification is known to alter histone function and chromatin architecture[34,35], enhance chaperone function[20], inhibit angiogenesis[21], potentiate necrotic cell death[36], and modulate neural precursor cell differentiation[37]. In particular, MGH-1 is known to be the most stable MGH isomer[8,14,38,39], and recent work has suggested that it, along with CEA, is among the most abundant cellular AGEs[34]. Thus, MGH-1 is thought to be one of the most prevalent and biologically relevant AGEs[8,9,19].

In this work, we use a combinatorial peptide library to determine the chemical features that favor MGH-1 formation for short peptides. We demonstrate that Tyr plays an active mechanistic role by accelerating the elimination that yields MGH-1, working cooperatively with a glutamate residue on the opposite peptide flank. We further demonstrate that these chemical features, which promote MGH-1 formation on short, unstructured peptides, can also predictably template selective glycation on full-length protein targets expressed in mammalian cells. The results herein contribute important knowledge about the landscape of glycation outcomes, including information about mechanistically related AGEs and the chemical features that control their formation in the absence of an enzyme. Such information will be critical for developing methodology that can be used to predict and/or control glycation in living cells.

## Results

Our previous work has revealed that primary sequence, on its own, shapes the distribution of resulting AGEs[29]. In our prior study, AGE products with an [M + 54] mass change, associated with formation of the MGH isomers, were the most frequently observed[29]. Of these, MGH-1 is known to be the most thermodynamically stable and has also been frequently detected in biological studies of glycation[8,14,34,38,39]. Thus, in this work, we aimed further to learn how primary sequence influences the formation of the specific AGE, MGH-1. Our study initiated with a panel of commercially available Arg-containing peptides that varied both in sequence composition and length. In as few as 3 h of MGO treatment, these peptides exhibited differences in total glycation (12–35% glycated), as assessed by liquid chromatography mass spectrometry (LC-MS). Importantly, levels of individual AGEs also varied substantially. In particular, the proportional levels of [M + 54] ranged from 8 to 35% of the total glycation observed for each peptide (Supplementary Figs. 1 and 2). Building on this observation, we designed a combinatorial library based on an Arg-containing sequence from human serum albumin (HSA) that we[29], and others[25], have found to be glycated in vitro (see Methods). After MGO treatment, sequences that produced MGH-1 were identified using a primary antibody specific for MGH-1 and subsequent colorimetric detection (Fig. 2a, b, Supplementary Figs. 3 and 4). Using this approach, 75 total "hit" beads were selected, cleaved and sequenced using LC-MS/MS with de novo sequencing (Supplementary Fig. 5). Because the strong basic conditions used to cleave peptides from the HMBA-linked resin certainly influence AGE distributions, this strategy was used simply to identify peptide sequences based only on the unmodified peptides remaining on each 'hit' bead.

Of the 75 beads selected, 55 unique sequences were identified; 11 were selected more than once and three were selected more than three times (Supplementary Table 1). Glycated sequences

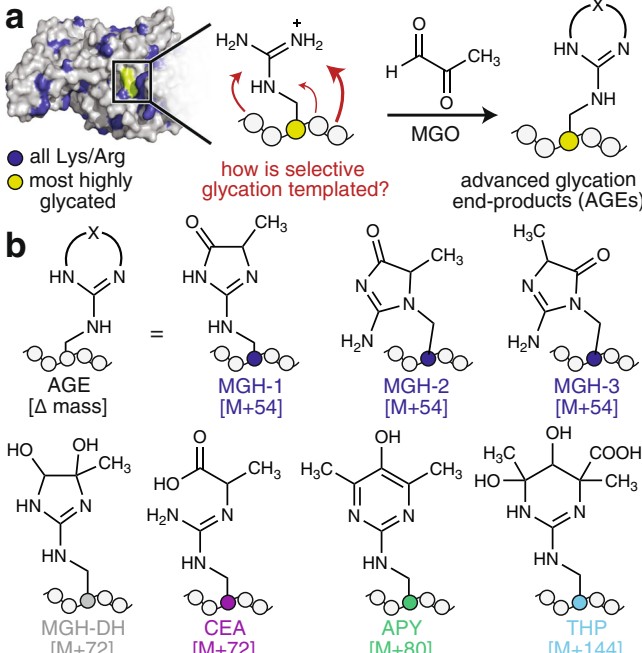

**Fig. 1 Glycation is a selective, nonenzymatic PTM. a** Glycation is a nonenzymatic post-translational modification (PTM) that occurs preferentially at certain Arg or Lys residues on certain proteins. For instance, human serum albumin (HSA, PDBID:1E7I) possesses 87 possible glycation sites (blue). Only 12 of these are reported to be glycated, and one site (green), is reported to be the most reactive. Yet, it remains an open question as to how the surrounding microenvironment—a combination of sequence and structure—templates the exact glycation outcome. For instance, the reaction of Arg with methylglyoxal (MGO), one of the most potent and prevalent cellular glycating agents, can produce numerous advanced glycation end-products (AGEs). **b** These include the methylglyoxal-derived hydroimidazolone isomers (MGH-1,-2,-3), dihydroxyimidazolidine (MGH-DH), carboxyethylarginine (CEA), argpyrimidine (APY), and tetrahydropyrimidine (THP).

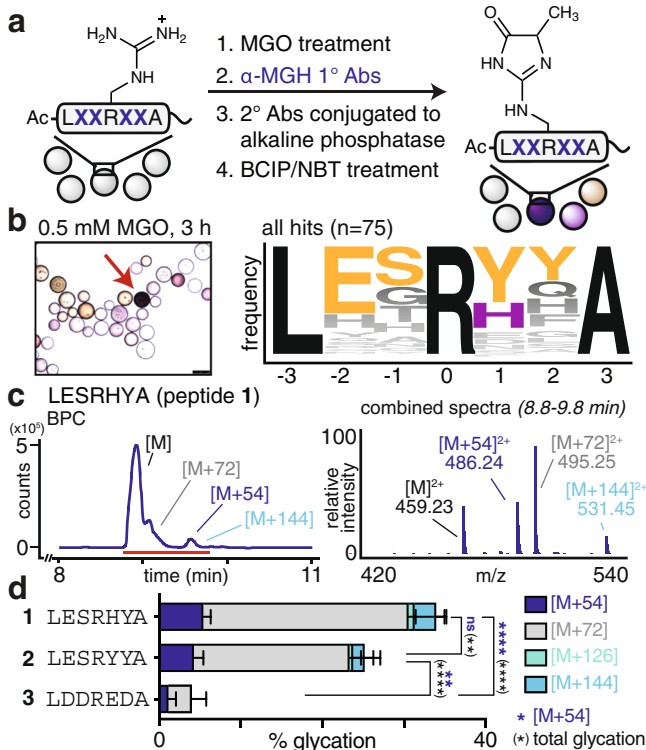

**Fig. 2 A combinatorial approach to identify peptides that favor a single MGH isomer. a** To identify sequences that promote MGH-1 formation, a library of peptides with randomized residues surrounding a central arginine were treated with 0.5 mM MGO for 3 h at 37 °C in 20 mM phosphate buffered saline (PBS) at pH 7.3. MGH-1-modified sequences were selected by incubation with α-MGH-1 primary antibodies, which was followed by treatment with secondary antibodies conjugated to alkaline phosphatase, and lastly, colorimetric detection with BCIP/NBT reagents. **b** Dark purple "hit" beads (red arrow) were selected, cleaved and identified using LC-MS/MS with de novo sequencing. 75 total hit beads were selected, sequenced, and found to converge on a consensus motif (LESRYYA, $n = 2$), which is nearly identical to the most frequently selected hit sequence (LESRHYA, $n = 6$). **c** Purified synthetic peptides that were N-terminally acetylated were allowed to react with equimolar MGO (1 mM) in pH 7.3 PBS for 3 h at 37 °C, and were then analyzed by LC-MS. Representative base peak chromatograms (BPC, left) and combined mass spectrum (right) for the reaction of the top hit, Ac-LESRHYA, (peptide **1**) with MGO. **d** Distribution of glycation products observed for the in vitro glycation of the most frequently selected hit sequence (Ac-LESRHYA, peptide **1**), the consensus motif (Ac-LESRYYA, peptide **2**) and a negative hit sequence, (Ac-LDDREDA, peptide **3**). Stacked bar graphs are plotted as mean ± standard deviation for each mass adduct. Individual data points are shown in Supplementary Fig. 8. Data are derived from independent experiments: peptide **1**, $n = 8$; peptide **2**, $n = 5$; peptide **3**, $n = 3$. Legend: blue, [M + 54]; gray, [M + 72]; green, [M + 126]; cyan, [M + 144]. A nondirectional (two-tailed) one-way ANOVA using Tukey's multiple comparison was used to determine if each peptide yielded statistically significant differences in [M + 54] (blue) or total glycation (black) levels. $p < 0.05$(*), $p < 0.01$(**), $p < 0.001$(***), $p < 0.0001$(****). Source data are provided as a Source Data file. Additional statistical information, including exact $p$-values, is available in an additional Supplementary Data file.

matching the epitopes used to generate α-MGH-1 antibodies were not identified (Supplementary Fig. 6). Instead, we found that the sequences converged on a consensus motif, which was also a hit (LESRYYA, ($n = 2$)). Although Tyr was the most frequently selected residue in both the +1 and +2 positions, the vast majority of hits displayed Tyr at only one of these sites (82% of

hits). Indeed, the consensus motif was the only hit to contain Tyr in both positions, while the most frequently selected hit (LESRHYA, ($n = 6$)) matches well with the consensus motif but replaces one Tyr with His (Fig. 2 and Supplementary Table 1).

To confirm that these selected sequences preferentially yielded MGH-1, we also screened using an alternative approach that distinguished hit beads using impaired proteolysis by trypsin, which cleaves the peptide backbone only when Arg remains unmodified (see Methods). This strategy selects glycated sequences that are independent of specific AGEs. We found that the consensus motif generated using this approach was different from that obtained when using α-MGH-1 antibodies (Supplementary Fig. 7). We also selected colorless beads exhibiting the lowest levels of glycation. The resulting sequences possessed a high density of acidic residues, corroborating our previous findings that clustered negative charge impairs glycation (Fig. 2, Supplementary Figs. 7 and 8, Supplementary Table 2)[29].

Our next goal was to confirm that sequences selected from the library reflected their intrinsic glycation susceptibilities. Thus, glycation was evaluated in solution for purified synthetic peptides matching the most frequently selected hit (Ac-LESRHYA, peptide **1**), the consensus motif (Ac-LESRYYA, peptide **2**), and a sequence identified from a colorless (nonglycated) bead (Ac-LDDREDA, peptide **3**). Peptide solutions (1 mM) were treated with MGO and the extent and type of glycation were quantified using LC-MS analysis. Mass adducts corresponding to unique AGEs were quantified relative to the remaining unmodified peptide, which we report as "% glycation" (See Supplementary Information). This quantification approach allows for a robust comparison of glycation extents across different AGEs on different peptide substrates, even though each may exhibit some variation in ionization efficiency[40]. We evaluated the glycation of peptides **1**, **2**, and **3** with MGO at 3 h and 24 h of treatment and found that the former is sufficient to reveal differences in MGH formation (Supplementary Fig. 8). Indeed, after 3 h of MGO treatment, peptides **1** and **2** yielded substantially higher levels of a single [M + 54] isomer (Fig. 2, Supplementary Fig. 8) whereas **3** led to low levels of glycation. These trends remained true after 24 h of treatment. However, with the extended incubation time, the formation of single addition products, like [M + 54], were overshadowed by large increases in MGO double additions, like [M + 144] (Supplementary Fig. 8).

Next, we aimed to determine the features present in peptide **1** that bias formation of a single [M + 54] adduct (Fig. 3a, b, Supplementary Fig. 9). We found that by replacing Glu in the −2 position with Gln (**1a**) there were no significant changes in glycation. In contrast, replacement with Asp (**1b**) led to a substantial decrease in both [M + 54] and total glycation levels. The substitution of Ser in the −1 position with Ala (**1c**) also led to minimal, but statistically significant, changes in [M + 54] and total glycation levels. Conversely, the Cys substitution (**1d**) led to notable increases in [M + 54] and the [M + 126] and [M + 144] double addition adducts (Fig. 3b). When Tyr in the +2 position was substituted with Phe (**1e**), a substantial decrease in [M + 54] levels was observed. This was also the case for the substitution of His in the +1 position with Phe (**1f**). Levels of total glycation and, in particular, [M + 54] were reduced even farther by double substitution of His and Tyr with Phe (**1g**). We obtained similar results for additional peptide **1** substitutions (**1h-k**, Supplementary Fig. 9) and a similar set of peptide **2** variants (**2a–2k**, Supplementary Fig. 10). These results suggest that Tyr plays a critical role in promoting [M + 54] adduct formation.

In addition to total glycation levels and AGE distributions, we also carefully monitored formation of isomeric AGEs. We found that replacement of Tyr with Phe (peptide **1e**) led to the formation of two distinct, chromatographically resolved, [M + 54]

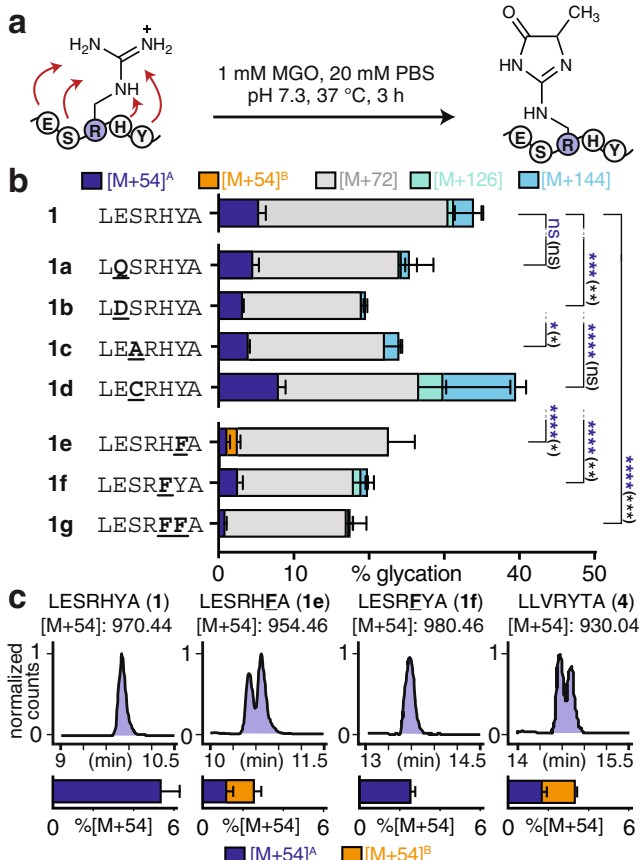

**Fig. 3 Tyrosine is essential to bias a single MGH isomer. a** To determine how each residue within the hit sequence contributes to glycation, point variants of peptide **1** were evaluated by LC-MS following in vitro glycation. Purified, N-terminally acetylated, synthetic peptides (peptides **1a–g**) were allowed to react with equimolar MGO (1 mM) in pH 7.3 PBS for 3 h at 37 °C. **b** Distribution of glycation adducts observed for variants of peptide **1** (**1**, $n = 8$; **1a-1g**, $n = 3$). Stacked bar graphs are plotted as mean ± standard deviation for each mass adduct. Individual data points are shown in Supplementary Fig. 9. Adduct legend: blue, $[M + 54]^A$; orange, $[M + 54]^B$; gray, $[M + 72]$; green, $[M + 126]$; cyan, $[M + 144]$. In general, substitutions made on the C-terminal peptide flank resulted in the greatest reduction in $[M + 54]$ and total glycation levels. A nondirectional (two-tailed) one-way ANOVA using Dunnett's multiple comparison test was used to determine if each variant yielded statistically significant differences in $[M + 54]$ (blue) or total glycation (black) levels compared to peptide **1**. $p < 0.05(*)$, $p < 0.01(**)$, $p < 0.001(***)$, $p < 0.0001(****)$. See also Supplementary Fig. 10. **c** Representative extracted compound chromatograms (ECCs) for $[M + 54]$ adducts reveal that a substitution in place of Tyr (peptide **1e**), but not His (peptide **1f**) results in the formation of multiple $[M + 54]$ isomers (i.e., MGH isomers). Multiple isomeric $[M + 54]$ products were also observed for a control sequence derived from HSA (peptide **4**), and for scrambled variants of peptide **1** (Supplementary Fig. 13). These results suggest that the specific placement of Tyr in peptide **1** is critical for the formation of a single $[M + 54]$ product. Source data are provided as a Source Data file. Additional statistical information, including exact p-values, is available in an additional Supplementary Data file.

isomers (Fig. 3c). In contrast, all other peptide **1** variants formed only a single isomer. We also found that a control sequence derived from HSA, which was also the basis for our library design (Ac-LLVRYTA, peptide **4**), was not selected during library screening and produced two distinct $[M + 54]$ isomers in roughly equal quantities (Fig. 3c, Supplementary Figs. 3 and 8). To confirm that this behavior is consistent when glycation is performed

on peptides displayed on resin, we completed a control experiment in which resin-bound peptides were subjected to the identical MGO and wash conditions used during library screening. For these experiments, we used a photocleavable linker to release peptides from the beads; this strategy avoids the use of strong base that is known to alter AGE distributions. This experiment revealed that, when resin-bound, peptide **1**, but not **4**, formed just a single MGH isomer and led to higher $[M + 54]$ levels (Supplementary Fig. 11). Together, these findings reconcile why peptide **1**, but not peptide **4**, was selected as a hit during library screening.

When Tyr was introduced at each position of a poly-Gly peptide, the highest levels of $[M + 54]$ were obtained when Tyr appeared in the +1 position (Supplementary Fig. 12). We also evaluated six scrambled peptide **1** variants, half of which led to multiple $[M + 54]$ isomers (Supplementary Fig. 13). These observations indicate that the presence, position, and relative placement of Tyr within peptide **1** are all essential to favor formation of a single $[M + 54]$ isomer (Fig. 3, Supplementary Figs. 8–13).

To determine the molecular role of Tyr in promoting formation of specific MGH isomers, we sought to understand the full scope of AGEs that form. Though peptide **1** was confirmed to favor a single $[M + 54]$ adduct, our initial studies also revealed that it is an $[M + 72]$ product ($[M + 72]^A$, $25.3 \pm 4.4\%$ modified), not $[M + 54]$ ($5.4 \pm 0.9\%$ modified), that dominates at early times (Figs. 2 and 3, Supplementary Fig. 8). During extended incubations, $[M + 72]^A$ is short-lived and negligible by 48 h (<1% modified) (Fig. 4a, b, Supplementary Fig. 14). This behavior is consistent with the dihydroxyimidazolidine (MGH-DH), a cyclic bis-hemiaminal[30,41,42]. We confirmed that $[M + 72]^A$ is MGH-DH using a chemical derivatization of vicinal diols by a boronic acid (Supplementary Fig. 15). Thus, the major AGE after 3 h of MGO treatment is an 'early' AGE that is either readily reversed or rearranged towards other, more stable, AGEs.

In contrast, the $[M + 54]$ adduct becomes the major product during extended incubations with MGO. We isolated the single peptide **1** $[M + 54]$ adduct and confirmed it to be MGH-1 by NMR (Supplementary Fig. 16). During the first 48 h, MGH-1 levels rise to $19.3 \pm 0.3\%$ modified and steadily increase to $26.2 \pm 1.5\%$ by 2 weeks and remain constant up to 4 weeks. Other products, like $[M + 144]$ (likely THP), also climb initially ($56.3 \pm 1.3\%$ modification at 72 h) but decrease dramatically thereafter ($11.8 \pm 0.8\%$ after 4 weeks). Thus, MGH-1 appears to have greater stability, as it becomes the major product only after other AGEs are depleted (Fig. 4a–c, Supplementary Fig. 14). We attribute this behavior to the fact that AGE formation likely occurs through a series of meta-stable intermediates that, if exposed to additional equivalents of MGO and/or solvent, continue to react, rearrange, or reverse.

Having established that AGE distributions change over time, we next sought to identify key mechanistic steps that could explain how Tyr promotes MGH-1 formation. To accomplish this goal, peptides were treated using our standard reaction conditions for 3 h and were then diluted 100× in the same buffer (Fig. 4a–c). By doing so, the formation of AGEs that require a second MGO equivalent is dramatically slowed. As a result, the intramolecular reactions and/or rearrangements of adducts existing at 3 h can proceed without competition, allowing us to connect the mechanistic dots between relevant intermediates. We found that MGH-DH decays rapidly upon dilution, dropping from $25.3 \pm 4.4\%$ to $13.3 \pm 4.8\%$ modified (49.6% loss) at 3 h post-dilution (hpd) and disappearing by 24 hpd. The loss in MGH-DH is concomitant with an increase in MGH-1, revealing MGH-DH as a direct precursor to MGH-1. We found that MGH-1 levels peak at 3 hpd ($13.88 \pm 1.22\%$ modified). However, somewhat

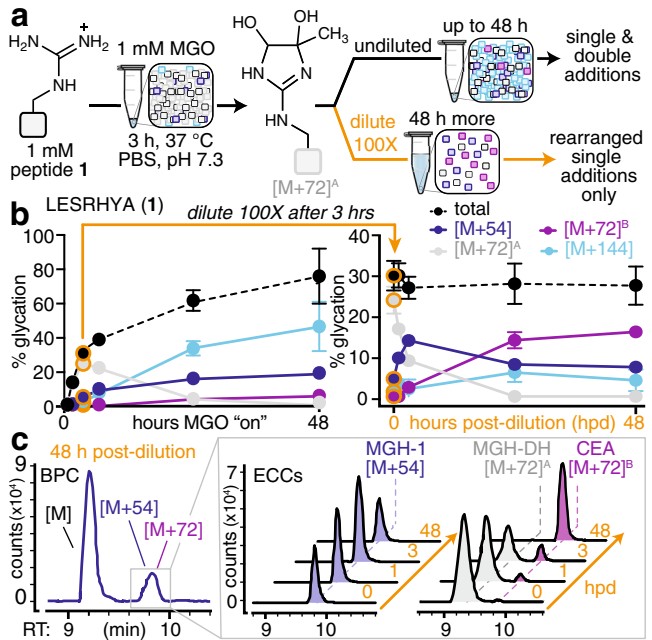

**Fig. 4 MGH-1 is mechanistically related to MGH-DH and CEA. a** To focus on the formation of single MGO additions, peptides were treated using our standard reaction conditions for 3 h and then diluted 100× in the same buffer. This adapted protocol slows the formation of AGEs that require a second MGO equivalent, but allows intramolecular reactions and/or rearrangements to proceed without competition. **b** Time course of glycation for peptide **1** treated with MGO for up to 48 h undiluted (left) or up to 48 h post-dilution (hpd) (right). After 3 h of treatment with equimolar (1 mM) MGO, the reaction was diluted 100× in the identical reaction buffer (orange arrows and data points). Time course data are plotted as mean ± standard deviation for each mass adduct. Legend: blue, [M + 54]; gray, [M + 72]$^A$; purple, [M + 72]$^B$; cyan, [M + 144]. Levels of [M + 126] remained at 5% or less throughout the entire time course and are not shown. **c** Representative base peak chromatogram (BPC) for peptide **1** at 48 h post-dilution (hpd). Extracted compound chromatograms (ECCs) of the [M + 54] (left) and [M + 72] (right) adducts observed from 0–48 h post-dilution (hpd). For experiments that confirm the identities of these adducts, please see Supplementary Figs. 15–18. Source data are provided as a Source Data file.

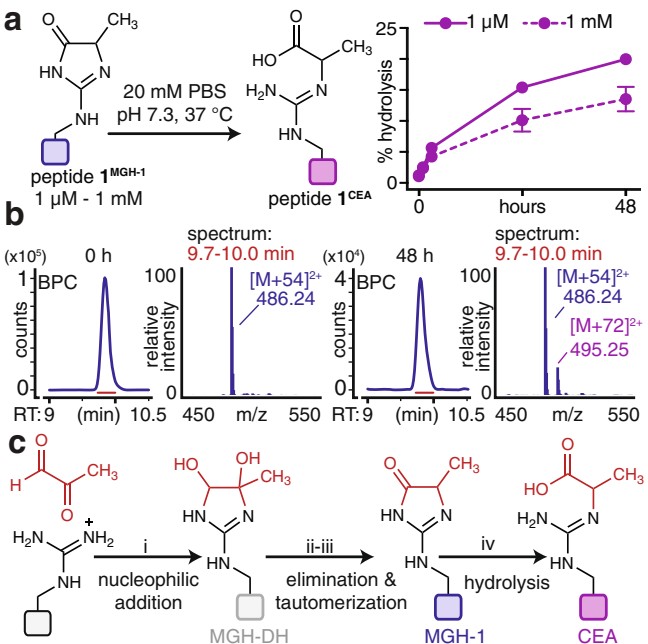

**Fig. 5 MGH-1 hydrolyzes to form CEA. a** Purified MGH-1-modified peptide **1** (peptide **1**$^{MGH-1}$, 1 μM–1 mM) was resuspended in phosphate buffered saline at pH 7.3 and incubated for up to 48 h at 37 °C. At all concentrations, peptide **1**$^{MGH-1}$ was found to hydrolyze directly to yield peptide **1**$^{CEA}$, as quantified by LC-MS. Data are plotted as mean values ± standard deviation, derived from independent hydrolysis experiments: 1 μM: 0 h ($n = 4$), 1 h ($n = 3$), 3 h ($n = 3$), 24 h ($n = 3$), 48 h ($n = 3$); 1 mM: 0 h ($n = 5$), 1 h ($n = 5$), 3 h ($n = 3$), 24 h ($n = 5$), 48 h ($n = 5$). See also Supplementary Fig. 19. **b** Representative BPCs for the incubation of pure peptide **1**$^{MGH-1}$ at 0 h and 48 h show the accumulation of peptide **1**$^{CEA}$ at later time points. We note that peptide **1**$^{MGH-1}$ and peptide **1**$^{CEA}$ elute closely in a single chromatographic peak. However, they have distinct mass changes that can be readily resolved by our analysis software, as shown in Fig. 4c. **c** Potential mechanistic steps involved in the formation of AGEs derived from a single addition of MGO. The first step is likely nucleophilic attack (i) of MGO by Arg to form the bis-hemiaminal MGH-DH. This intermediate undergoes elimination (ii) and subsequent tautomerization (iii) to form MGH-1. Our work further demonstrates that MGH-1 can be hydrolyzed (iv) to yield CEA. Source data are provided as a Source Data file.

unexpectedly, MGH-1 levels were subsequently found to decay between 6 and 24 hpd as a new, chromatographically resolved [M + 72]$^B$ isomer forms (Fig. 4c). By 48 hpd, this new [M + 72]$^B$ is the major product (16.00 ± 1.29% modified) (Fig. 4b, c). This product was unable to be derivatized by the boronic acid reagents used to detect MGH-DH, (Supplementary Fig. 15), and is therefore more consistent with the known AGE carboxyethylarginine (CEA) (Fig. 1). To determine the identity of this product, we isolated the peptide **1** [M + 72]$^B$ adduct and confirmed it to be CEA using NMR and chemical derivatization (Supplementary Figs. 17 and 18).

Although CEA is known to be a hydrolysis product of MGH-3[39,43–45], this has not been reported for MGH-1. In fact, MGH-1 has been reported to be resistant to hydrolysis, as past work has indicated that CEA must form through an MGH-3 intermediate[8,14,34,38,39]. To confirm that MGH-1 can be a precursor to CEA, we isolated pure MGH-1-modified peptide **1** (peptide **1**$^{MGH-1}$), and found that it generated CEA, on its own, when resuspended in buffer (Fig. 5a, b). Our studies spanned a range of low (1 μM–1 mM) peptide **1**$^{MGH-1}$ concentrations (Fig. 5a, Supplementary Fig. 19), allowing us to conclude that the extent of CEA formation was highest when lower concentrations

of peptide **1**$^{MGH-1}$ were used. This reconciles past observations that MGH-1 is recalcitrant to hydrolysis, as prior studies were performed at high MGH-1 (up to 6 mM[39]) concentrations and/or using amino acid monomers that cannot capture the influence of surrounding functional groups[34,38,39,46]. Additionally, using forcing, basic conditions (pH 12), we were able to identify a new peptide **1** [M + 54] adduct with a distinct retention time from peptide **1**$^{MGH-1}$, likely corresponding to MGH-3. We purified this product and found that the rate of its hydrolysis to CEA was greatly enhanced relative to the observed rate of hydrolysis for peptide **1**$^{MGH-1}$ at all concentrations tested (63.15–70.77% glycated, Supplementary Fig. 20). Furthermore, this putative MGH-3 adduct appeared more susceptible to the reversal of glycation, as an appreciable amount (21.51–27.52%) of unmodified peptide **1** was regenerated during the 48 h of incubation. This behavior was not observed while performing comparable studies for peptide **1**$^{MGH-1}$ (Fig. 5a, b & Supplementary Fig. 19). Given that the profile of CEA formation observed in our dilution studies (Fig. 4b), where AGE-modified peptide concentrations are low, tracks very closely with that which is observed for peptide **1**$^{MGH-1}$ (Fig. 5a, b and Supplementary Fig. 19) rather than the putative MGH-3-modified peptide **1** adduct (Supplementary Fig. 20),

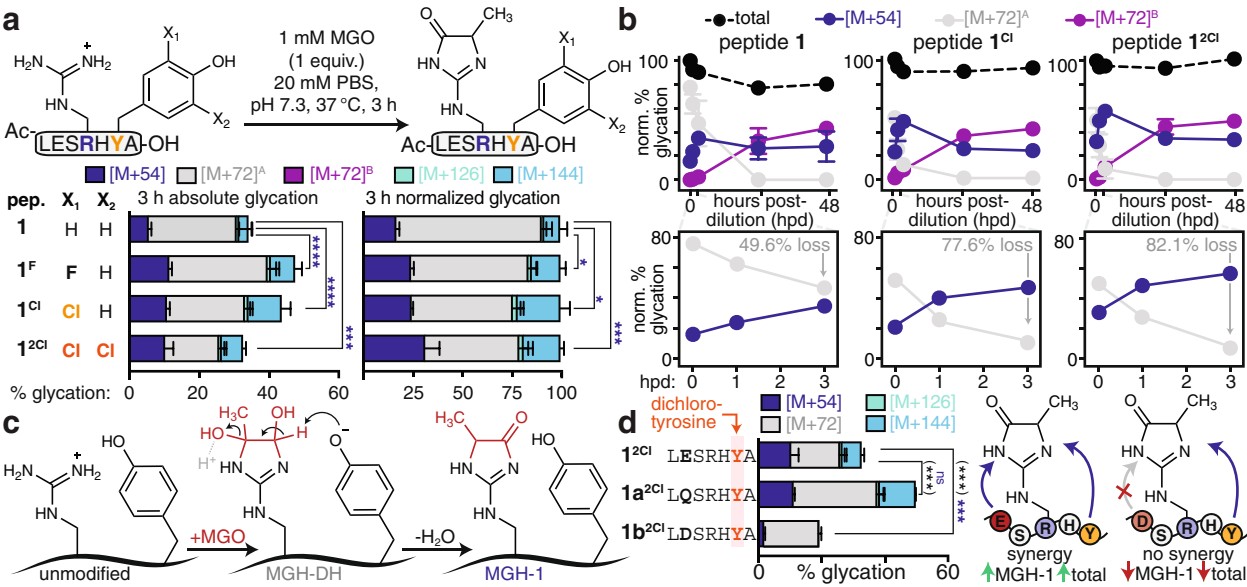

**Fig. 6 Tyr plays an active mechanistic role and works cooperatively with Glu to favor MGH-1. a** The role of the tyrosine phenol was evaluated by replacing Tyr ($pK_a = 10$) with 3-fluoro-Tyr ($pK_a = 8.2$, **1^F**), 3-chloro-Tyr ($pK_a = 8.1$, **1^Cl**), or 3,5-dichloro-Tyr ($pK_a = 6.2$, **1^2Cl**) and then performing in vitro glycation. Absolute (left) and proportional (right) distribution of AGEs. Stacked bar graphs are plotted as mean values ± standard deviation, derived from independent experiments: peptide **1**, $n = 8$; **1^F**, $n = 3$; **1^Cl**, $n = 3$; **1^2Cl**, $n = 3$. Legend: blue, [M + 54]; gray, [M + 72]^A; purple, [M + 72]^B; green, [M + 126]; cyan, [M + 144]. Individual data points are shown in Supplementary Fig. 21. **b** Glycation adducts observed for peptides **1**, **1^Cl**, and **1^2Cl** upon dilution after 3 h of MGO treatment. The total amount of glycation observed at 3 h was normalized to 100% for each peptide. As seen in the first 3 h post-dilution (hpd), the conversion of MGH-DH to MGH-1 occurs faster for peptides **1^Cl** and **1^2Cl**. Data plotted are mean values ± standard deviation, derived from independent MGO time course experiments. Legend: blue, [M + 54]; gray, [M + 72]^A; purple, [M + 72]^B; black, total glycation. For clarity, the [M + 126] and [M + 144] adducts were removed from **b**, as they remain constant throughout the time studied. For full time course studies, please see Supplementary Fig. 23. **c** These data are consistent with a model in which Tyr acts as a general base, accelerating the elimination of MGH-DH, with water as the likely leaving group, to form MGH-1. **d** Distribution of glycation adducts observed for variants of peptide **1^2Cl** that replace Glu with Gln (**1a^2Cl**) or Asp (**1b^2Cl**). Stacked bar graphs are plotted as mean ± standard deviation; peptide **1^2Cl**, $n = 3$; **1a^2Cl**, $n = 4$; **1^2Cl**, $n = 3$. Individual data points are plotted in Supplementary Fig. 21. A nondirectional (two-tailed) one-way ANOVA using Dunnett's multiple comparison was used to determine if each variant yielded statistically significant differences in [M + 54] (blue) compared to **a** peptide **1** or **d** peptide **1^2Cl**. $p < 0.05$(*), $p < 0.01$(**), $p < 0.001$(***), $p < 0.0001$(****). Source data are provided as a Source Data file. Additional statistical information, including exact $p$-values, is available in an additional Supplementary Data file.

these results lend further support to our conclusion that peptide **1^MGH-1** is directly hydrolyzed to peptide **1^CEA**. Thus, our results are most consistent with a model in which MGH-DH is an intermediate that gives rise to MGH-1, which itself can be considered an intermediate leading to CEA formation at low MGO and peptide concentrations (Fig. 5c).

Building on these observations, we considered that the Tyr phenol could have a mechanistic role that would favor MGH-1 formation in peptide **1**. To evaluate this hypothesis, we synthesized peptide **1** variants that replaced Tyr ($pK_a = 10$) with the unnatural derivatives 3-chloro-Tyr ($pK_a = 8.1$, peptide **1^Cl**) and 3,5-dichloro-Tyr ($pK_a = 6.2$, peptide **1^2Cl**) (Fig. 6a, Supplementary Figs. 21 and 22). As compared to **1**, peptides **1^Cl** and **1^2Cl** exhibit far greater levels of the Tyr phenoxide at near-neutral pH. After 3 h of MGO treatment, peptides **1^Cl** and **1^2Cl** yielded not only greater absolute levels of MGH-1 than peptide **1**, but also higher proportional levels of MGH-1, which became apparent after normalizing total glycation levels to 100% (Fig. 6a, Supplementary Fig. 21). We also prepared a fluorinated variant, peptide **1^F** ($pK_a = 8.2$), which has a nearly identical phenolic $pK_a$ and was glycated comparably to **1^Cl**. For peptide **1^2Cl**, there was less total glycation as compared to **1^Cl**, though absolute MGH-1 levels remained similar. At pH 7.3, the majority of peptide **1^2Cl** carries three discrete negative charges, which we have previously shown to impair glycation[29]. Thus, the decrease in total glycation for peptide **1^2Cl** is primarily due to lesser MGH-DH formation, suggesting that dense negative charge impedes initial MGO addition, not the subsequent elimination yielding MGH-1. We

also compared the glycation of peptides **1**, **1^Cl**, and **1^2Cl** at pH 6 and 12 and found that AGE distributions were similar when each peptide has the same charge (Supplementary Fig. 22). Moreover, after 48 h of MGO treatment, peptide **1^2Cl** yielded the highest levels of MGH-1 (Supplementary Fig. 23). Thus, these results suggest that the Tyr phenoxide is critical for promoting MGH-1 formation.

Next, we sought to determine if the Tyr phenoxide could facilitate the, presumably rate-determining, elimination/dehydration that converts MGH-DH to MGH-1. We evaluated the transformation of MGH-DH to MGH-1 for peptides **1**, **1^Cl**, and **1^2Cl** by diluting the reaction after 3 h of MGO treatment. In all cases, the most rapid disappearance of MGH-DH and simultaneous appearance of MGH-1 occurred within the first 3 hpd (Fig. 6b). The rate of MGH-DH disappearance was accelerated for peptides **1^Cl** and **1^2Cl** (77.6% and 82.1% losses at 3 hpd), whereas all three peptides exhibited roughly similar AGE distributions at 48 hpd (Fig. 6b, Supplementary Fig. 23). These results suggest that the Tyr phenoxide acts as a general base that facilitates elimination of MGH-DH to yield MGH-1 (Fig. 6c).

We also found that decreasing the phenolic $pK_a$ stabilized glycation levels over time (Fig. 6b). In peptide **1**, glycation drops by 19.3% over 48 hpd, which we attribute to the reversal of glycation. By decreasing the phenolic $pK_a$ in peptides **1^Cl** and **1^2Cl** and favoring the elimination path yielding the more stable MGH-1, glycation levels drop only 7.2% for peptide **1^Cl** and remains constant for **1^2Cl** (Fig. 6b, Supplementary Fig. 23). This observation suggests that the rate increase for MGH-DH elimination to

form MGH-1 outcompetes reversal to liberate unmodified peptide for **1**$^{Cl}$ and **1**$^{2Cl}$. Similarly, peptides that are not optimized for MGH-1 formation, including **1e**, **1f**, and **4**, exhibited an even greater decrease in glycation over 48 hpd (29.1%, 31.0%, and 27.1% respectively) (Supplementary Fig. 24). These findings suggest that a properly tuned microenvironment modulates glycation by extending AGE lifetimes.

Given that Tyr alone is not sufficient to favor MGH-1, we further sought to identify synergistic interactions within peptide **1**. In particular, we focused on a potential medium-range interaction between Tyr and Glu in the +2 and −2 positions, respectively, as substitutions at these sites led to significant changes in glycation (Fig. 3b). Moreover, this placement of Glu was the most common co-occurrence for library hits with Tyr in the +2 position (Supplementary Table 1). When Glu was replaced with Gln in peptide **1**$^{2Cl}$ (peptide **1a**$^{2Cl}$) there was virtually no change in MGH-1 levels after 3 h of MGO treatment, though total glycation increased (Fig. 6d). We attribute this change to the reduction of negative charge, which increases overall glycation but, notably, does not alter MGH-1 levels. We suspect that MGH-1 levels remain roughly constant because Gln is able to recapitulate polar contacts also made by Glu. In contrast, when Glu is replaced by Asp (peptide **1b**$^{2Cl}$), there is a dramatic reduction in both MGH-1 and total glycation levels. This change led to an 85% reduction in MGH-1, while total glycation was reduced by only 42% (Fig. 6d). In this case, the loss of a side chain methylene appears to sacrifice necessary, cooperative interactions that promote MGH-1 formation. This change is more pronounced for chlorinated peptide **1** variants that further favor MGH-1, suggesting that Glu and Tyr at opposite ends of the peptide work together to facilitate MGH-1 formation. These results reveal that an individual negative charge optimally positioned in space can enhance glycation through cooperative effects with nearby residues that facilitate chemical transformations to form stable AGEs.

Numerous past studies have relied on proteomics to identify cellular targets of glycation, and, in some cases, biologically pertinent glycation events[34,35,47–51]. These studies are well suited for cataloging AGE-modified cellular proteins[48–51], but have so far been unable to uncover the underlying chemical features that govern preferential glycation[29]. By contrast, our studies herein have revealed how the surrounding chemical environment—in this case approximated by changes in sequence—directs specific glycation outcomes. In particular, our results have discerned distinct mechanistic steps and cooperative interactions that influence the formation of discrete AGEs, like MGH-1, for short, unstructured peptides in vitro. Thus, our next goal was to evaluate if these findings are directly applicable to the glycation of full-length proteins expressed in living mammalian cells.

Prior work, including our own, has unambiguously demonstrated that there is no consensus sequence for glycation[29,48–51]. Therefore, the use of unbiased cell-based proteomics or targeted studies of known glycated proteins would be ineffective at reporting on the sequence effects we sought to test. Instead, we designed an experiment to evaluate explicitly if the sequence effects that govern the glycation of peptides in vitro remain relevant for a full-length protein in a cellular environment. In particular, the results of our in vitro studies confirmed that peptide **1** is glycated to a greater extent than peptide **3** and also forms significantly greater levels of an [M + 54] adduct corresponding to MGH-1. To evaluate if such findings are meaningful for understanding how glycation is controlled on protein substrates and in cells, we expressed green fluorescent protein (GFP) variants that were C-terminally fused either to the peptide **1** sequence (-LESRHYA, GFP-**1**), the peptide **3** sequence (-LDDREDA, GFP-**3**), or a glycation-inert sequence lacking a

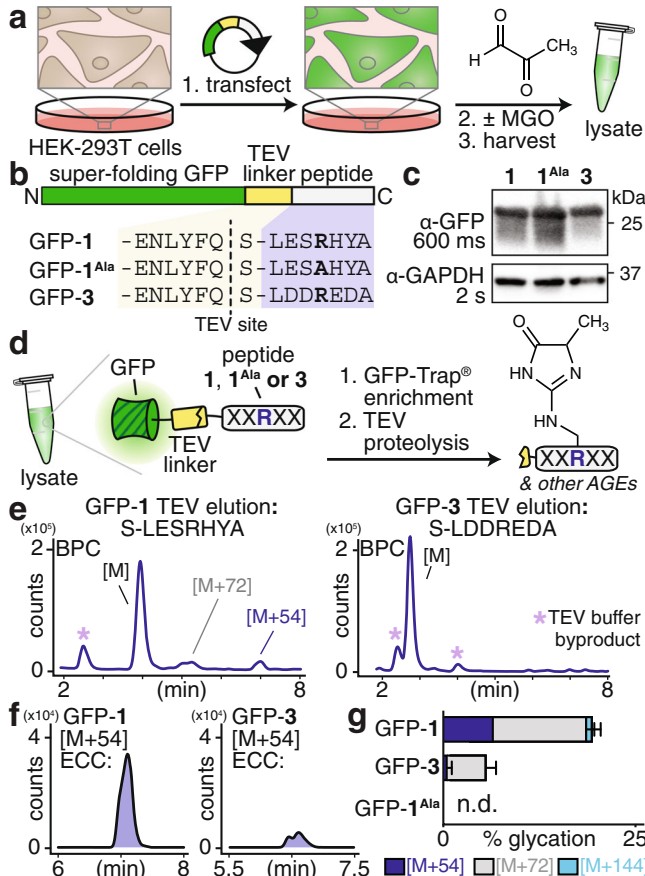

**Fig. 7 Templating selective glycation for full-length proteins in mammalian cells. a**, **b** Plasmids encoding the green fluorescent protein (GFP) fused to peptides **1**, **1**$^{Ala}$, or **3** linked via the tobacco etch virus (TEV) protease recognition site were transiently transfected into HEK-293T cells and subsequently treated with or without MGO. After treatment, cells were harvested and lysed. **c** Western blot analysis, probing with α-GFP antibodies, revealed that each GFP variant was expressed comparably in in HEK-293T cells. **d** After lysis, lysates were subjected to GFP enrichment using the commercially available GFP-Trap® agarose resin, followed by incubation with TEV protease to liberate the C-terminal peptides (**1**, **1**$^{Ala}$, or **3**). The resulting peptides were analyzed by LC-MS. **e** Base peak chromatograms (BPC) for peptides released from GFP-**1** (left) and GFP-**3** (right) following treatment with 5 mM MGO and GFP enrichment. Peaks corresponding to AGEs were only apparent in the peptide released from GFP-**1**. Byproducts of the TEV protease reaction buffer are indicated (*, light purple). **f** Extracted compound chromatograms (ECC) for the [M + 54] adduct reveal that its formation was substantially enhanced for the peptide released from GFP-**1** (left) but not GFP-**3** (right). **g** Distribution of glycation products observed for each of the peptides released from GFP-**1**, GFP-**3**, and GFP-**1**$^{Ala}$ following MGO treatment (5 mM) and GFP enrichment/TEV cleavage, based on two replicate experiments. Legend: blue, [M + 54]; gray, [M + 72]; cyan, [M + 144]. Individual data points are shown in Supplementary Fig. 27. Source data are provided as a Source Data file.

central Arg (-LESAHYA, GFP-**1**$^{Ala}$) in HEK-293T cells (Fig. 7a). These C-terminal peptide sequences were connected to GFP through a linker sequence containing a tobacco etch virus (TEV) protease cleavage site (Fig. 7b). We confirmed that all variants were equivalently expressed in HEK-293T cells after 24 h of transient transfection (Fig. 7c and Supplementary Figs. 25 and 26). Next, cells were exposed to MGO for 3 h and then harvested and lysed. To monitor the glycation of GFP-**1**, GFP-**3**, and GFP-**1**$^{Ala}$, we immunoprecipitated against GFP, and then used TEV

protease to liberate the C-terminal peptide **1**, **3**, or **1^Ala** fragment from GFP (Fig. 7d, Supplementary Fig. 26). The resulting eluates were analyzed by LC-MS to determine the extent and type of glycation (Fig. 7e–g, Supplementary Fig. 27). We found that only GFP-**1** led to appreciable formation of an [M + 54] product, likely corresponding to MGH-1. C-terminal peptide fragments from GFP-**1** also led to higher levels of an [M + 72] adduct, which could be either MGH-DH or CEA, and small quantities of [M + 144]. In contrast, GFP-**3** yielded only vanishingly small [M + 54] levels and GFP-**1^Ala** led to undetectable [M + 54] levels (Fig. 7f–g; Supplementary Figs. 26 and 27). This result is consistent with our initial peptide studies (Fig. 2) and remained true even when the MGO concentration was titrated (Supplementary Fig. 26). Collectively, these results demonstrate that the chemical features that guide selective glycation for peptide substrates in vitro are directly applicable for full-length protein targets in cellular systems.

## Discussion

Herein we have taken an unbiased combinatorial approach that has revealed glycation to be controlled by the surrounding chemical microenvironment and guided by MGO concentration. Past mechanistic studies have focused primarily on single amino acid derivatives and/or monomers[39,43,44,52], which were unable to capture critical long- and medium-range interactions that we have shown not only to bias the distribution of resulting AGEs but also to play an active mechanistic role in their formation. In particular, this study has defined a critical role for Tyr in promoting MGH-1 formation. Although Tyr has previously been proposed to be important for glycation, its exact role had not been determined experimentally[25]. Our results further reveal that Tyr acts as a general base, promoting the elimination of water from MGH-DH to yield MGH-1. As a result, our study is the first to reveal specific mechanistic steps connecting multiple AGEs, including MGH-DH, MGH-1, and CEA. We also demonstrate that CEA is a hydrolysis product of MGH-1 and provide evidence that both peptide and MGO concentrations influence glycation outcomes. The work herein has further uncovered that selective glycation is driven by factors other than Arg $pK_a$, conflicting with prior reports[25–28,53]. In particular, peptides possessing chlorinated Tyr, with lowered phenolic $pK_a$'s, likely elevate the nearby Arg $pK_a$. Yet, these peptides yielded the most MGH-1, suggesting that decreased Arg $pK_a$ is not prerequisite for MGH-1 formation. Instead, our results clarify that selective glycation and formation of specific AGEs is instead governed by nearby polar and/or ionizable groups that facilitate rate-determining mechanistic steps, such as the elimination of MGH-DH to form MGH-1 in peptide **1**. Thus, our findings not only provide critical insight about mechanistically connected AGEs, but also further suggest that certain sequences, and/or microenvironments, confer susceptibility to promote not just hydroimidazolone formation, but also the specific type of MGH isomer.

Multiple proteomics studies have reported that clustered acidic residues are overrepresented surrounding glycated sites[48,54,55]. Yet, our findings unambiguously show that clustered negative charge impedes glycation, most likely the initial addition of MGO. Our results further uncover that that acidic residues, when properly oriented, participate in cooperative interactions that are critical to form MGH-1. Our work has revealed that not only the identities, but also the arrangements and interactions of residues are essential to the glycation outcome. These observations reconcile why acidic residues appear frequently in proteomic data sets, but on their own are poor predictors for glycation[53,54,56]. It is currently unclear if Glu might interact directly with Tyr to favor MGH-1, or if they each exert a mechanistic influence through interactions with MGO-bound Arg. Our future work will focus on distinguishing these two possibilities, as well as on applying this approach to understand the features that control the formation of other AGEs.

For folded proteins, well-organized microenvironments are scaffolded not only by primary sequence, but also by secondary, tertiary, and quaternary structure. In this case, the exact AGE outcome may be influenced by involvement from distal residues that, when properly folded, are placed in close proximity to the reactive Arg. Here we demonstrated that short, unstructured peptides are able to mimic these effects, both in vitro and in living mammalian cells, by providing a distinct, if more loosely defined, microenvironment that influences glycation at the central Arg. Our work therefore provides clear evidence that glycation is influenced through medium-, or potentially long-range interactions within short, unstructured peptides that may translate to larger protein targets. The experiment we performed in mammalian cells, in which peptide **1** or **3** was fused to GFP, allowed us to test only if the sequence trends seen in vitro were relevant to glycation of intact proteins by MGO. By doing so, we avoided conflating multiple, distinct questions regarding the separate roles of sequence and structure in influencing cellular glycation outcomes. Our future work will focus on evaluating how the sequence effects identified herein are affected by secondary or tertiary structure, $T_m$, or other properties of folded proteins. Such information will be particularly useful for predicting likely glycation sites on fully, folded native proteins. Unraveling these molecular rules is crucial for developing new methods that can be used to control AGE formation in living cells and will enable the study of glycation as a functional post-translational modification.

## Methods

**General materials**. All chemical reagents and solvents were of analytical grade, obtained from commercial suppliers and used without further purification unless otherwise noted. Methylglyoxal (40% w/v in water) was purchased from MilliporeSigma. For complete Materials & Methods, please refer to the Supplementary Information.

**Liquid chromatography mass spectrometry analysis and quantification**. Reversed phase liquid chromatography and mass spectrometry (LC-MS) analysis was carried out using an Agilent 1260 Infinity LC system in-line with an Agilent 6530 Accurate Mass Q-TOF. Peptide reaction mixtures were injected onto an AdvanceBio Peptide 2.7 μm column (2.1 × 150 mm, Agilent). MS analysis was accomplished using Agilent MassHunter BioConfirm Qualitative Analysis software and PEAKS Studio (v. 7.5) software. MS data was quantified using the MassHunter Molecular Feature Extractor, which reports cumulative MS ion counts as 'volumes' observed for any and all charge states associated with a particular ion. Quantification (% modification) was carried out by dividing the specified mass adduct(s) volume by the total volume of modified and unmodified peptide. Retention times were used to identify discrete isomers with degenerate masses.

**General protocol for glycation of peptides**. To perform in vitro glycation, to 30 μL of ultrapure water was added 5 μL of a 10 mM stock peptide solution in 50% DMF/water, 10 μL of 100 mM phosphate buffered saline (PBS) at pH 7.3 and, lastly, 5 μL of a 10 mM MGO stock in water. The final concentrations were 1 mM peptide and 1 mM MGO in 20 mM PBS with 5% DMF co-solvent. Tubes were capped, briefly spun in a benchtop microcentrifuge and incubated in a 37 °C water bath, typically for 3 h and in some cases for up to 4 weeks. After incubation, peptide samples were diluted (1:100) into 5 mM Tris at pH 7.3 to quench the reaction, unless otherwise noted, and subsequently subjected to LC-MS analysis. For dilution experiments, after 3 h of MGO incubation, a 10 μL reaction aliquot was diluted into 990 μL of 20 mM PBS at pH 7.3 and incubated at 37 °C for up to 48 additional hours, then directly subjected to LC-MS analysis.

**Library design and synthesis**. The library scaffold ((Ac/Bio)-Ahx-LXXRXXA-A_βA_βA_βA_β) consisted of a C-terminal tetra-β-alanine spacer, a central peptide sequence, an amino-hexanoic acid (Ahx) linker, and, finally, an N-terminal cap (either biotin (Bio) or acetyl (Ac)). The four variable positions were randomized with all of the canonical amino acids (Ala, Asn, Asp, Gln, Glu, Gly, His, Leu, Phe, Ser, Thr, Trp, Tyr, and Val), except for those with degenerate masses (Ile), prone to oxidation (Cys and Met), susceptible to competition with the central arginine (Arg and Lys), or exhibiting potentially confounding structural effects (Pro). The

resulting combinatorial library contained 38,416 sequences conjugated C-terminally to the aqueous compatible ChemMatrix® resin (100–200 mesh, 0.50 mmol/g loading) via an HMBA linker. Standard solid phase peptide chemistry was used to achieve the library synthesis (see Supplementary Information).

**Library screening with α-MGH-1 antibodies**. For the complete protocol, please refer to the Supplementary Information. Briefly, 120 mg of N-terminally acetylated peptide library was equilibrated in DI water overnight. The next day, beads were exposed to 0.5 mM MGO for 3 h at 37 °C. Following incubation, excess MGO was removed and the resulting MGO-modified beads were blocked using a 1% solution of bovine serum albumin. Next, the blocked beads were incubated with a 1:1000 dilution of α-MGH-1 antibody (Cell BioLabs, Inc.) for 18–20 h. After washing, beads were then exposed to a 1:1000 dilution of α-mouse secondary antibody (Abcam) conjugated to alkaline phosphatase for 4 h. Following washing and equilibration in alkaline phosphatase buffer, the beads were exposed to color developing reagents 5-bromo-4-chloro-3-indolyl phosphate and nitro blue tetrazolium (BCIP/NBT, Promega) and were transferred to a petri dish and imaged using brightfield microscopy. Roughly 120,000 beads were screened in three replicate experiments. Based on our library size, this sample size ensures that our library was appropriately sampled (see Supplementary Information). A total of 75 dark purple beads were manually selected and cleaved with 100 mM sodium hydroxide. The resulting single-bead cleavage mixtures were diluted 50% in 100 mM hydrochloric acid to neutralize the solution, filtered using fritted micro-centrifuge spin columns and subjected directly to LC-MS/MS analysis.

**Library screening using impaired proteolysis by trypsin**. For the complete protocol, please refer to the Supplementary Information. Briefly, 120 mg of N-terminally biotinylated peptide library was equilibrated in DI water overnight. The next day, beads were exposed to 0.5 mM MGO for 3 h at 37 °C. Following incubation, excess MGO was removed and the resulting MGO-modified beads were exposed to sequencing-grade trypsin (Promega) for 48 h. After trypsin exposure, beads were washed and then blocked, as described above. Beads were then incubated with a 1:1000 dilution of streptavidin alkaline phosphatase (Promega). Following washing and equilibration in alkaline phosphatase buffer, the beads were exposed to color developing reagents, imaged, manually selected, cleaved, and subjected to LC-MS/MS analysis, as described above.

**Cellular MGO assays**. For experiments to evaluate glycation of GFP variants, roughly 1 million HEK-293T cells were seeded in a 10 cm sterile tissue culture dish and grown for 18–24 h. Dishes were transfected with 10 µg of the desired plasmid using TransIT-LT1 Transfection Reagent (Mirus Bio). After transfection, cells were cultured for an additional 24 h before MGO treatment (2.5 or 5 mM final concentration) in DMEM) at 37 °C at 5% $CO_2$ for up to 3 h. After MGO treatment, cells were harvested with TrypLE Express (Gibco). Harvested cells were transferred to a 15 mL conical tube, and pelleted at $200 \times g$. The resulting pellet was washed with 3–5 mL of 20 mM PBS, pH 7.3 and stored on ice until lysis. Cells were lysed on ice in 400–600 µL Tris-Cl buffer (50 mM Tris, 150 mM NaCl, 1 mM EDTA, 1 mM NaF, 1% Triton X-100) at pH 7.5 with a Pierce Protease and Phosphatase Inhibitor tablet (1 tablet/10 mL buffer). Lysates were clarified by centrifugation at $4255 \times g$ for 30 min, and total protein quantified by BCA Protein Assay (Pierce) on a Tecan Spark 10 M plate reader and analyzed by western blot (10 µg/sample) or used for immunoprecipitation.

**Reporting summary**. Further information on research design is available in the Nature Research Reporting Summary linked to this article.

## Data availability

Additional statistical information, including exact $p$-values, are available as an additional Supplementary Data file. All other results supporting the findings of this study are available within the paper and its supplementary information files. Additional data are available from the corresponding author upon reasonable request. Source data are provided with this paper.

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

## Acknowledgements

This work was supported in part by the National Institute of General Medical Sciences of the National Institutes of Health under award number R01 GM132422. This work was also supported by the Smith Family Award for Excellence in Biomedical Research, the Natalie V. Zucker Award, and a Tufts Faculty Research Fund Award to R.A.S., as well as the Poole Summer Fellowship to J.M.M. We gratefully acknowledge Prof. Bob Stolow and Dr. Xiangjin Song for assistance with NMR, and J. Kritzer, K. Kumar, and C. Bennett, and members of the Scheck lab for helpful comments during the preparation of the manuscript.

## Author contributions

S.F. performed the library design, optimization, screening, and initial validation studies. J.D. performed studies of glycation on commercially available peptides. A.S.G. performed peptide synthesis for scrambled and poly-Gly sequences and evaluated their performance in in vitro glycation assays. J.M.M. performed further validation studies, as well as all time course assays, dilution assays, mechanistic studies, chemical derivatization, characterization of AGEs by NMR, studies to define synergistic interactions that influence AGE formation, cellular MGO assays, and assisted in the preparation of the figures and manuscript. R.A.S. designed the project, directed experimental design and analysis, and prepared the figures and manuscript.

## Competing interests

The authors declare no competing interests.
