## [Peer Review File · Nature Communications]

Reviewers' Comments:

Reviewer #1:

Remarks to the Author:

The area of non-enzymatic glycation is an important research area with potential to make huge impact in health and disease as the authors of this manuscript described in the introduction and background of the study. ... et. al. made a commendable attempt to address an interesting question "the factors involved in selectivity for arginine by methylglyoxal in glycation of proteins or in this study in peptide library". The aim of this study is to understand formation of mechanistically-related AGEs and the chemical features that control their formation in the absence of an enzyme. To enhance methodology that can be used to predict and/or control glycation in living cell. However, there have been many studies, providing key insights in this area of research studying whole protein molecules (34) under physiological condition not the peptides as in this study. To identify a methylglyoxal (MG) glycation motif, authors used combinatorial peptide library to determine the chemical features that favour MG-H1. It is a detailed study to learn how primary sequence influences the formation of MGH-1. Unfortunately the study contain fundamental flaws. Major weakness is that the pre-analytical processes compromised the validity of the outcome.

Major concerns

Methodological flaws

1. Use of commercial highly impure methylglyoxal, have many other reactive compound could affect the outcome. This is surprising because it is well established that the commercial MG is highly impure and method for preparation of high impurity MG has been published in a well-known protocol (1).

2. Authors were aware of the problem of using strong basic conditions to cleave glycated peptides from the HMBA-linked resin that harsh conditions (100 mM NaOH) used to cleave the beads degrade MG-H1 please see (2). The effect of high pH compromise this study on 2 counts; 1) it profoundly degrades MG-H1, 2) it causes the MG moiety of residual MG-H1 to move between the residues, compromising MG modification motif identification (3).

3. In this study, authors did not provide any analytical recoveries of the methods employed herein. They need to show the MG-H1 levels, before and after the cleavage, were the same for both methods (UV cleavage and the High pH cleavage). This can only be done by exhaustive enzymatic hydrolysis of the glycated and un-glycated peptide.

4. Formation of CEA appears to be approximately linear i.e. proportionate with time from the start of the incubation and not dependent on MG-H1 precursor (Fig 4. MGH-1 is Mechanistically Related to MGH-DH and CEA). Previous studies of the reaction of MG with protein shows stability of MG-H1 out to 21 days (1). This inconsistency suggest CEA may be formed from a contaminant in the crude MG used. The study needs to be repeated with high purity MG.

Conceptual flaws

1. Using peptide library to understand glycation process in primary structure in physiological system is flawed because the effects of secondary and tertiary structure for spatial neighbouring group is lost. This is crucial in directing sites of MG modification see for example reference 34 in the manuscript (4). Therefore, study design is not appropriate and needs major revision

2. It is not possible to model microscopic pKa changes of arginine with short peptides because these are influenced by secondary and tertiary structure of proteins; cf. requirement of protein crystal structure information to compute microscopic pKa of arginine and other ionisable amino acid residues (3)

Minor concerns

Throughout the manuscript the description and some figures of experimental outcome is confused and do not provide accurate experimental finding. For example, MG-H1 rearrange to form CEA. Not clear if all peptides were glycated individually or combined.

What are the analytical recoveries of resin cleavage in high pH (100 mM NaOH) as half-life of MG-H1 is extremely low at high pH.

MGO-modified beads were blocked using a 1% solution of bovine serum albumin (BSA). BSA is not a suitable blocking agent to study AGEs using antibodies because commercially available BSA is

glycated and may compromise the results

Reference

1. Biochem. J. 364, 15 – 24, 2002.
2. Nature Protocols 9, 1969 – 1979, 2014
3. Glycoconjugate J. 33, 553-568 (2016)
4. J. Biol. Chem. 280, 5724–5732 (2005) (Ref 34 in the manuscript).

Reviewer #2:

Remarks to the Author:

McEwen and co-workers provide a valuable, in-depth, and detailed study to pursue an important question in glycochemistry – why do some positions in proteins undergo particular glycation reactions, relative to similar side chain functional groups at other positions in the protein sequence?

This question is addressed by monitoring the outcome of glycation reactions conducted on a library of peptide substrates immobilized on solid support and exposed to methylglyoxal. Peptides that undergo glycation preferentially are identified by immunochemical techniques, using an antibody specific to the presence of a particular methylglyoxal-derived hydroimidazolone isomer, MGH-1. Different glycation products are then identified and quantified using LC-MS techniques. The study is successful in identifying putative neighboring sequence characteristics that pre-dispose a particular arginine residue to undergo specific glycation reactions. In addition, the study provides a valuable analysis of mechanistic issues regarding the transformation/maturation of different glycation products.

The report is recommended for acceptance in Nature Communications following some minor revisions, as follows.

The experimental approach implies that one particular glycation product is particularly critical – that of MGH-1, and the study protocols and the manuscript's discussion reflect that emphasis. The justification for the focus on MGH-1, relative to other MGH isomers, or other products such as "CEA", "APY", and "THP" should be explicitly clarified for the reader.

The characteristics of the antibody used should be evaluated – to what extent does the antibody recognize MGH-1 as an epitope, relative to other glycation products?

75 "hit" beads were chosen for analysis. How many beads were used in this experiment? What proportion of the input beads were stained by the immunochemical techniques? What type of analysis was conducted to establish that characterization of 75 beads was sufficient to rigorously identify the consensus motif?

Interpretation of results should be reconsidered – in some cases, seemingly subtle differences in reaction outcomes are associated with significant differences in glycation susceptibility. For example, in Figure 3a, peptide sequence 1a (a Glu to Gln modification) was described as not yielding a significant change in glycation outcome, but peptide 1c (a Ser to Ala modification) was described as giving rise to diminished glycation products. However, casual inspection of the data in Figure 3a does not seem to provide clear evidence for any significant difference in comparing these two peptides.

Identification of particular glycation products associated with particular mass values may be excessively speculative. It would be best if the authors merely reported their observations regarding the relative abundance of particular mass species, and then in a separate section provided their supposition regarding the likely prevalence and interconversions of particular glycation products. More speculative discussion and the generation of hypotheses regarding mechanism could be better separated from the presentation of the data obtained.

The text does not make clear whether relative intensity of mass peaks in the LC-MS experiments is strictly correlated with relative concentration of the particular glycation products in solution. The

extent to which this was ascertained or calibrated should be clarified.

Several of the figures are exceedingly "busy", and may try to provide too much information in too compact a space. It is therefore quite difficult for the reader to interpret some of these figures, which may be a particular problem for the broad readership of Nature Communications, who may not be previously familiar with glycochemistry. Particular glycation products are color-coded in the figures (e.g., Fig. 4B), but it is often necessary to cross-reference these color codes with previous figures in order to properly interpret the data. It's fairly daunting, particularly when confronted with abbreviations in small font in the graph axes such as "hpd" (hours post dilution?).

Reviewer #3:

Remarks to the Author:

This is a very well-written manuscript that sheds light on the site-specificity of non-enzymatic protein glycation events, a topic that is rarely considered or discussed. The figures are very well done and provide relatively clear results. While I believe this manuscript would be of interest to the Nature Communications readers, I think that a few major points need to be addressed. Providing an NMR of MG-H1 and MG-H3 is critical, as the authors claim that CEA can be generated from MG-H1 without providing this direct evidence (more on this below). In addition, I have following concerns:

1. Why was the 3h timepoint chosen? Presumably shorter, or longer, incubations would result in a different profile of modifications. For example, the generation of CEA requires an intermediate (see DOI: 10.1021/ja301994d), so the full spectrum of modifications may be missed with this incubation period.

2. As discussed and shown in Figure 3c, two distinct MG-H isomers can be observed, with MG-H1 being the predominate species. The mechanism shown in Figure 4e may not be correct: as shown by Galligan et al. (DOI: 10.1073/pnas.1802901115) and Wang et al. (see DOI: 10.1021/ja301994d), CEA results from the hydrolysis MG-H3, while MG-H1 remains (largely) stable. Did the authors confirm their findings via NMR? Further, if I am interpreting the figures correctly, it appears that the substitution of Phe at position +2 is sufficient to drive the formation of MG-H3, while other substitutions here do not. This would further push the formation of CEA, which is what is observed in 1e, 1f, and 1g, where the modifications are largely limited to CEA, rather than MG-H1. If this is true, this is perhaps equally as interesting as these findings now not only confer the relatively susceptibility of a given sequence to MG-H modification, but also the type of MG-H (and ultimately CEA) modification.

3. Can the authors speculate on the role of tertiary structure in conferring Arg susceptibility to MGO modification?

4. The authors tend to focus on the type of modification being generated, which is obviously a major component of this manuscript; however, which peptide yielded the highest total amount of glycation. The graphs tend to show most peptides yield ~10-40% glycation, whereas some 6a-d are above 50%. I think this is an important consideration.

5. Confirmation of these findings on an intact protein would significantly increase the validity of these findings. Certainly, a complete proteomic survey of glycation sites is beyond the scope of this manuscript; however, can the authors identify sites on recombinant proteins with similar treatment conditions? A small survey of known glycated proteins would presumably yield sites with similar primary sequences as having the highest degree of modification. Further, this may yield interesting information on the susceptibility of Arg residues on a folded protein, rather than just focusing on the primary sequence.

6. The evidence supporting the authors claim that "We also demonstrate, for the first time, that CEA is a hydrolysis product of MGH-1, and provide evidence that MGO concentrations influence glycation outcomes." Is not entirely convincing. This has been shown to not be the case, again, see DOI: 10.1021/ja301994d. To demonstrate this point, the authors will need to chemically

synthesize MG-H1 and conduct a similar "hpd" experiment and quantify CEA. Based on in vivo measurements of MGO glycation profiles, it appears CEA is the most abundant modification (DOI: 10.1073/pnas.1802901115), with MG-H1 being second. Under these long-term conditions, should MG-H1 result in CEA, very little MG-H1 would be observed.

Response to Reviewers

Reviewer #1

“The area of non-enzymatic glycation is an important research area with potential to make huge impact in health and disease as the authors of this manuscript described in the introduction and background of the study. ... et. al. made a commendable attempt to address an interesting question “the factors involved in selectivity for arginine by methylglyoxal in glycation of proteins or in this study in peptide library”. The aim of this study is to understand formation of mechanistically-related AGEs and the chemical features that control their formation in the absence of an enzyme. To enhance methodology that can be used to predict and/or control glycation in living cell. However, there have been many studies, providing key insights in this area of research studying whole protein molecules (34) under physiological condition not the peptides as in this study. To identify a methylglyoxal (MG) glycation motif, authors used combinatorial peptide library to determine the chemical features that favour MG-H1. It is a detailed study to learn how primary sequence influences the formation of MGH-1. Unfortunately the study contain fundamental flaws. Major weakness is that the pre-analytical processes compromised the validity of the outcome.”

We appreciate Reviewer 1's feedback and agree that the question we seek to address herein is an interesting and important one. Reviewer 1 points out that *“there have been many studies, providing key insights in this area of research studying whole protein molecules (34) under physiological condition not the peptides as in this study.”* We fully recognize that there have been many past efforts to identify the features that guide selective glycation on protein substrates, and this prior work has established that glycation occurs selectively for many proteins.¹⁻²⁴ However, those studies have rationalized findings individually without further experimental validation. For example, in ref. 34, mentioned by Reviewer 1, the Thornalley lab reported preferential glycation of Arg410 in human serum albumin (HSA) by MGO. Based on an analogy to the HSA esterase mechanism, the authors proposed that MGO was activated by an initial hemiacetal with a nearby Tyr, thereby facilitating attachment of MGO to Arg410. However, this proposal was not validated experimentally. This type of approach, which is pervasive in the glycation literature,¹⁻²⁴ has led to many *ideas* about features that might promote glycation but few, if any, have been confirmed. Moreover, each study has been limited in scope and/or performed using conditions that are not directly comparable. Thus, these remain isolated reports that do not provide a unified understanding of the features that promote selective glycation.

In contrast, our previously published work (ref. 38 in the manuscript),²⁵ and our work reported herein, are the first, to our knowledge, to experimentally validate guidelines for selective glycation that have the potential to be generalized. By using an *in vitro* approach and focusing on the glycation of peptides, our strategy offers a practical, innovative alternative to past work by enabling us to identify features that promote the formation of specific AGE-products. Moreover, compared to cell-based proteomics, or even *in vitro* studies using full-length proteins, our approach simplifies the identification of *any*, not only expected, AGEs, many of which are isomeric and can be difficult to discern in a proteomics workflow. This enables us to evaluate the fundamental question about the underlying chemical features that promote glycation, rather than simply enumerating cellular or *in vitro* glycation events. These advantages provide an unprecedented view of glycation under highly controlled conditions, which is most relevant to understand not only the intrinsic propensity of a given site to be glycated, but also the chemical mechanisms through which specific AGEs may form. In this work, we capitalize on these advantages to reveal the chemical features that promote formation of MGH-1, one of the most abundant and biologically-relevant AGEs.

We strongly disagree with the characterization that any of the points raised by Reviewer 1 are *“fundamental flaws”* and, as we will describe in detail below, we have thoroughly addressed the concern that *“the pre-analytical processes compromised the validity of the outcome.”* Our responses are addressed on a point-by-point basis below:

Response to Reviewers

1. Use of commercial highly impure methylglyoxal, have many other reactive compound could affect the outcome. This is surprising because it is well established that the commercial MG is highly impure and method for preparation of high impurity MG has been published in a well-known protocol (1).

We appreciate Reviewer 1's suggestion that a well-known protocol for distilling high purity MGO has been previously described, though we question if such a protocol would add much value in our study. Indeed, 1,2-dicarbonyls, particularly those that, like MGO, possess enolizable carbons, are notoriously prone to polymerization even after distillation. Although our experiments do, of course, vary from day to day (the degree to which is reflected in the error bars in our figures, as well as in the source data provided in our revised submission), we have not noticed any major differences that result from different bottles of commercial MGO or that are introduced over extended storage using the manufacturer recommended conditions. Thus, we suspect that the level of impurity in commercial MGO is either fairly constant and/or easily reversed once administered in our experiments. Moreover, our study seeks to provide useful information about how selective glycation is templated in biological systems. In a biological context, MGO is certainly not pure, as it is just one of many biologically-relevant aldehydes (not to mention metabolites in general) that each has the potential to cross-react and/or form side products in addition to the known AGEs. Thus, adding a step to further purify MGO seems, to us, irrelevant in this context. Indeed, numerous recent publications in this journal^{26,27} and other top journals²⁸ have performed well-controlled biological studies using the same commercial source of MGO, and we have published our prior work in *Angewandte Chemie*²⁵ using these same materials.

That being said, we appreciate Reviewer 1's concern that the use of commercial MGO could affect our results in such a way that would diminish their applicability to glycation in a biological system. We felt that the most direct way to address this concern was to perform a new experiment that would allow us to determine if our conclusions remained applicable in living mammalian cells. Briefly, we expressed green fluorescent protein (GFP) fusions that were C-terminally fused to the peptide **1** sequence (-LESRHYA, GFP-**1**), the peptide **3** sequence (-LDDREDA, GFP-**3**), or a negative control sequence inert to glycation (-LESAHYA, GFP-**1**^{Ala}) in HEK-293T cells. These C-terminal peptide sequences were connected to GFP through a linker sequence containing a tobacco etch virus (TEV) protease cleavage site. To monitor the glycation of GFP-**1**, GFP-**3**, and GFP-**1**^{Ala}, we treated cells with MGO following standard protocols.²⁹ After cells were harvested and lysed, we immunoprecipitated against GFP and then used TEV protease to liberate the C-terminal peptide (**1**, **3**, or **1**^{Ala}) fragment from GFP. Using LC-MS, we found that only GFP-**1** led to appreciable formation of an [M+54] adduct (likely MGH-1), as well as [M+72] adducts, whereas GFP-**3** yielded only vanishingly small quantities of [M+54] and GFP-**1**^{Ala} led to undetectable [M+54] levels. ***These new results can be found in a newly added Fig. 6, with further data in new Supplementary Figs 24 & 25, along with additional discussion in the manuscript.*** Together, these data convincingly demonstrate that the chemical features we identified, which guide selective glycation in peptide substrates, are also relevant to the glycation of full-length protein targets expressed in living mammalian cells. Therefore, there is no cause for concern that the use of commercial MGO has in any way diminished the outcome or the impact of our results.

2. Authors were aware of the problem of using strong basic conditions to cleave glycated peptides from the HMBA-linked resin that harsh conditions (100 mM NaOH) used to cleave the beads degrade MG-H1 please see (2). The effect of high pH compromise this study on 2 counts; 1) it profoundly degrades MG-H1, 2) it causes the MG moiety of residual MG-H1 to move between the residues, compromising MG modification motif identification (3).

We agree with Reviewer 1 that the strongly basic conditions used to cleave peptides from HMBA-linked beads can influence the complement of AGEs observed by LC-MS. We were aware of this issue, which is why we did not quantify the extent of glycation for resin-bound peptides in this study.

Response to Reviewers

Instead, after exposure of our resin-bound peptide library to MGO and subsequent selection and detection, we used strong base simply to liberate peptides from the resin. We could then identify the unique peptide sequence by performing MS/MS on the remaining unmodified peptide. Thus, our selection strategy was agnostic to the specific AGEs that might be detected after exposure to strong base. Additionally, each member of the library had only one Arg residue (and no Lys) and a capped N-terminus, so there was only one possible site of glycation. The subsequent quantification reported in our study was performed for experiments where purified, synthetic peptides matching the “hit” sequences were treated with MGO in solution, and their glycation was assessed using LC-MS. **We have added new text to the main text and Supplementary Information to clarify this point**, and thank Reviewer 1 for letting us know that this critical aspect of our study was not clearly stated in our original submission.

3. *In this study, authors did not provide any analytical recoveries of the methods employed herein. They need to show the MG-H1 levels, before and after the cleavage, were the same for both methods (UV cleavage and the High pH cleavage). This can only be done by exhaustive enzymatic hydrolysis of the glycated and un-glycated peptide.*

As previously explained (see *comment #2*, above), [M+54] levels reported in the main text were quantified only for glycation reactions in solution, so determining analytical recovery does not appear to be relevant to the experiments that were performed. Similarly, the purpose of the experiment that involved UV-assisted cleavage of resin-bound peptides (described in Supplementary Fig. 11) was to assess if our findings from the aforementioned solutions studies matched with the results of glycation that was performed on-resin. Thus, we used a photocleavage to liberate glycated peptides from the resin, which avoids many of the pitfalls associated with the use of strong base. Indeed, using this strategy, we were able to demonstrate that—both in solution and on-resin—peptide **1** (LESRHYA) leads to higher [M+54] levels than peptide **4** (LLVRYTA), and forms a single [M+54] adduct. **We have added new text to the main text of the manuscript to clarify this point** and we note that determination of the analytical recovery would not impact this conclusion.

4. *Formation of CEA appears to be approximately linear i.e. proportionate with time from the start of the incubation and not dependent on MG-H1 precursor (Fig 4. MGH-1 is Mechanistically Related to MGH-DH and CEA). Previous studies of the reaction of MG with protein shows stability of MG-H1 out to 21 days (1). This inconsistency suggest CEA may be formed from a contaminant in the crude MG used. The study needs to be repeated with high purity MG.*

We agree with Reviewer 1 that past work has suggested that CEA cannot be formed directly from MGH-1. To unambiguously determine that MGH-1 is indeed a direct precursor to CEA, we purified MGH-1-modified peptide **1** and confirmed it to be MGH-1 using NMR (peptide **1**^{MGH-1}) (Supplementary Fig. 16). We then incubated pure peptide **1**^{MGH-1} in buffer for up to 48 hours and analyzed the reaction using LC-MS. We found that peptide **1**^{MGH-1} is hydrolyzed directly to result in the formation of the peptide **1** [M+72]^B adduct (Fig. 4c-e). Although those findings were reported in the supplementary materials from our prior submission, we performed a new and improved version of this experiment using different concentrations of purified peptide **1**^{MGH-1} (Fig. 4d, Supplementary Fig. 19). We found that the extent of hydrolysis was greater when lower concentrations of peptide **1**^{MGH-1} were used. **These new data can be found in main text Figure 4 and Supplementary Fig. 19, along with additional discussion in the main text of the manuscript.** We also unequivocally confirmed that the peptide **1** [M+72]^B adduct is peptide **1**^{CEA} using NMR and chemical derivatization. **These new data appear in Supplementary Fig. 17 and Supplementary Fig. 18.** Because CEA forms from a pure sample of verified peptide **1**^{MGH-1}, our data are most consistent with a model in which MGH-1 is directly converted to CEA.

Using forcing conditions to prepare larger quantities of peptide **1**^{MGH-1}, we also were able to identify a small peptide **1** [M+54] peak (most likely MGH-3) with a distinct retention time from peptide **1**^{MGH-1}.

Response to Reviewers

We purified this product and found that the rate of its hydrolysis to CEA was greatly enhanced relative to the observed rate of hydrolysis for peptide **1**^{MGH-1}. ***This new data can be found in Supplementary Fig. 20.*** Given that the profile of CEA formation observed in our dilution studies, where AGE-modified peptide concentrations are low, tracks very closely with that which is observed for peptide **1**^{MGH-1} (Fig. 4 and Supplementary Fig. 19) rather than the putative MGH-3-modified peptide **1** adduct (Supplementary Fig. 20), these results lend further support to our conclusion that peptide **1**^{MGH-1} is directly hydrolyzed to peptide **1**^{CEA}, despite prior findings to the contrary by other groups.

5. *Using peptide library to understand glycation process in primary structure in physiological system is flawed because the effects of secondary and tertiary structure for spatial neighbouring group is lost. This is crucial in directing sites of MG modification see for example reference 34 in the manuscript (4). Therefore, study design is not appropriate and needs major revision*

Our prior study,²⁵ published a few years ago in *Angewandte Chemie*, found that selective glycation is not correlated with a lowered Arg pK_a or with increased solvent exposure, as suggested in past work.^{2,30–33} Instead, our previously published work has revealed that primary sequence is a major driver for glycation.²⁵ Primary sequence dictates both the overall susceptibility for a site to become glycated and also influences the specific distribution of AGEs that form at that site. In this work, we demonstrate this point even more definitively by discerning the specific ways in which primary sequence influences the formation of discrete AGEs, like MGH-1. In particular, our combinatorial peptide-based approach reveals that, when properly positioned, tyrosine plays an active mechanistic role that facilitates MGH-1 formation. Moreover, we have demonstrated that—even within short, unstructured peptides—nearby side chain functional groups work cooperatively to promote formation of specific AGEs (see Fig. 3, Fig. 5 & Supplementary Fig 23). Thus, while past work by others has considered sequence and structure together, as mentioned by Reviewer 1, our work provides clear evidence that sequence alone can contribute to the specific AGE outcome. This finding has practical implications, because it allows for the study of glycation using simpler substrates, like peptides, that can deconvolute sequence effects from structural effects. This finding also has an important conceptual implication, because it enables us to reveal (and experimentally validate) the molecular features that govern the intrinsic propensity of a certain site to form certain AGEs.

Thus, we agree with Reviewer 1 that our study removes the effects of “*secondary and tertiary structure*”. However, we see this as an advantage that simplifies the system and allows us to make more robust conclusions than would be possible from protein substrates, like the ones we used in our prior study.²⁵ However, we are still able to generate information about the influence of “*spatial neighbouring groups*”, as 7-mer peptides certainly present a distinct molecular environment surrounding the reactive Arg, which influences its reactivity. We show that primary sequence can govern the overall propensity for a site to become glycated, as well as influence the resulting distribution of AGEs that form at that site, as evidenced by the distinct sequence motifs derived using different library screening methods (Fig. 2 & Supplementary Figs. 7 & 8). We further demonstrate this to be true even for peptides that vary by only a single amino acid substitution or by a different order of the same residues (Figs. 3 & 5 and Supplementary Figs. 9, 10, & 13). Moreover, as seen in a newly added Fig. 6, the trends that influence glycation on short peptides *in vitro* remain applicable for full-length protein substrates expressed in mammalian cells. Therefore, our careful experimental design and our resulting conclusions make clear that the study design is both appropriate and novel for this field.

6. *It is not possible to model microscopic pKa changes of arginine with short peptides because these are influenced by secondary and tertiary structure of proteins; cf. requirement of protein crystal structure information to compute microscopic pKa of arginine and other ionisable amino acid residues (3)*

Response to Reviewers

We thank Reviewer 1 for this comment, and we agree that there have been many studies, including the one mentioned, that calculate Arg pK_a perturbations using available structural information. However, in this study, we did not model or compute any microscopic pK_a changes based on secondary or tertiary structure. It appears this comment is in response to our discussion about how the Arg pK_a in peptide **1** would be expected to be influenced by a nearby Tyr when it is substituted with mono- and di-chloro Tyr derivatives. Using the Henderson-Hasselbalch equation, it is straightforward to conclude that, with a pK_a of 6.8, the majority of di-chloro Tyr will be deprotonated with a reaction pH at 7.3. In this case, a proximal negative charge (introduced on peptide **1**^{2Cl}) can be expected to raise the nearby Arg pK_a due to a stabilization of the positively-charged Arg conjugate acid. This is a fundamental principle of acidity that is taught in most Biochemistry courses, particularly those with a high degree of chemical rigor. As a result, Arg pK_a in this context (peptide **1** vs. peptide **1**^{2Cl}) can be predicted based solely on chemistry fundamentals, and without requiring any computation or modeling. Our discussion is in complete agreement with such a treatment.

7. Throughout the manuscript the description and some figures of experimental outcome is confused and do not provide accurate experimental finding. For example, MG-H1 rearrange to form CEA.

This point was addressed in our response to *comment #4* from Reviewer 1. We have performed rigorous, new experiments to even more strongly demonstrate that MGH-1 can indeed hydrolyze to form CEA. ***These new data are found in main text Fig. 4, Supplementary Fig. 17, Supplementary Fig. 18, Supplementary Fig. 19, and Supplementary Fig. 20. We have also revised the text significantly,*** which we hope will provide greater clarity about our experimental methods and our conclusions.

8. Not clear if all peptides were glycosylated individually or combined.

We apologize for any lack of clarity about our methods in the original submission. This information was previously described in the *Methods* section, with even more detail available in the *Supplementary Information*. However, in this revision, ***we have added new text to the Results section and Supplementary Information to clarify the experimental methods used.***

9. What are the analytical recoveries of resin cleavage in high pH (100 mM NaOH) as half-life of MG-H1 is extremely low at high pH.

As mentioned in our response to *comment #3*, we did not quantify MGH-1 levels after treatment with 100 mM NaOH, so it was not necessary to determine analytical recoveries.

10. MGO-modified beads were blocked using a 1% solution of bovine serum albumin (BSA). BSA is not a suitable blocking agent to study AGEs using antibodies because commercially available BSA is glycosylated and may compromise the results.

We are grateful to Reviewer 1 for pointing this out. During library screening, all beads were treated with this blocking solution. Thus, even if there were a possibility for low-level background signal from the small proportion of glycosylated BSA, the detection strategy was still able to amplify true signal from only the most modified beads. We performed many controls to optimize our library screening conditions (which can be found in Supplementary Fig. 3, Supplementary Fig. 4, and Supplementary Fig. 5). Furthermore, the hit sequences identified from our library were then independently and rigorously validated using solution studies that did not include detection reagents or blocking agents such as BSA (as seen in Fig. 2 and Supplementary Fig. 8). Thus, in our view, there is no concern that the use of this reagent could have compromised our results.

Response to Reviewers

Reviewer #2

“McEwen and co-workers provide a valuable, in-depth, and detailed study to pursue an important question in glycochemistry – why do some positions in proteins undergo particular glycation reactions, relative to similar side chain functional groups at other positions in the protein sequence?”

This question is addressed by monitoring the outcome of glycation reactions conducted on a library of peptide substrates immobilized on solid support and exposed to methylglyoxal. Peptides that undergo glycation preferentially are identified by immunochemical techniques, using an antibody specific to the presence of a particular methylglyoxal-derived hydroimidazolone isomer, MGH-1. Different glycation products are then identified and quantified using LC-MS techniques. The study is successful in identifying putative neighboring sequence characteristics that pre-dispose a particular arginine residue to undergo specific glycation reactions. In addition, the study provides a valuable analysis of mechanistic issues regarding the transformation/maturation of different glycation products.”

The report is recommended for acceptance in Nature Communications following some minor revisions, as follows.

We greatly appreciate this feedback from Reviewer 2. A point-by-point response to their specific comments can be found below:

1. The experimental approach implies that one particular glycation product is particularly critical – that of MGH-1, and the study protocols and the manuscript’s discussion reflect that emphasis. The justification for the focus on MGH-1, relative to other MGH isomers, or other products such as “CEA”, “APY”, and “THP” should be explicitly clarified for the reader.

We thank Reviewer 2 for pointing this out. ***We have added text to the Introduction as well as the beginning of the Results section*** to further clarify that we chose to focus on MGH-1 as it is one of the most abundant AGEs and is also thought to be one of the most biologically-relevant. It was also the AGE that was most frequently detected in our prior study, making it an excellent starting point for the current work.

2. The characteristics of the antibody used should be evaluated – to what extent does the antibody recognize MGH-1 as an epitope, relative to other glycation products?

We appreciate this comment from Reviewer 2, and in response ***we have added additional text to the Supplementary Information***, which provides more detail about the antibody used. Briefly, the α -MGH-1 antibody was originally reported in *Molecular Medicine* in 2002.³⁴ The antibody was developed using MGH-modified ovalbumin. It has been reported to be specific for MGH-1, based on a competition assay using several different AGE-modified antigens, and it was also validated to detect MGH-1 using synthetic MGH-1-modified immunogens.³⁵

3. 75 “hit” beads were chosen for analysis. How many beads were used in this experiment? What proportion of the input beads were stained by the immunochemical techniques? What type of analysis was conducted to establish that characterization of 75 beads was sufficient to rigorously identify the consensus motif?

We are grateful to Reviewer 2 for pointing this out, and we apologize that more information about our library protocol was not included in our original submission. In the revised manuscript, ***we have added additional text to the Methods section and the Supplementary Information***, which describes that roughly 360,000 (120,000 x 3 replicates) beads were screened for a library of close to 40,000 unique sequences. From these 360,000 beads, all beads that produced a dark purple color were selected and sequenced, yielding just 75 hits. These protocols are based on widely accepted standards for sampling one-bead one-compound peptide libraries.³⁶

Response to Reviewers

4. Interpretation of results should be reconsidered – in some cases, seemingly subtle differences in reaction outcomes are associated with significant differences in glycation susceptibility. For example, in Figure 3a, peptide sequence 1a (a Glu to Gln modification) was described as not yielding a significant change in glycation outcome, but peptide 1c (a Ser to Ala modification) was described as giving rise to diminished glycation products. However, casual inspection of the data in Figure 3a does not seem to provide clear evidence for any significant difference in comparing these two peptides.

We appreciate this point from Reviewer 2. We apologize for the lack of consistency in our prior submission, which likely was a result of a split focus (on our part) in describing both the magnitude of glycation differences and the statistical significance for each while discussing the results in Fig. 3. In practice, however, we did indeed focus our attention on substitutions that produced *both* noticeable changes in AGE levels/distributions *and* were statistically significant. **The manuscript text has now been revised to reflect this approach more accurately.**

5. Identification of particular glycation products associated with particular mass values may be excessively speculative. It would be best if the authors merely reported their observations regarding the relative abundance of particular mass species, and then in a separate section provided their supposition regarding the likely prevalence and interconversions of particular glycation products. More speculative discussion and the generation of hypotheses regarding mechanism could be better separated from the presentation of the data obtained.

We greatly appreciate this comment and we have revised the manuscript accordingly. We agree that this is an important change, and we were pleased to find that this revision significantly improved the flow of our narrative. Specifically, we found that by reporting the observations about relative abundances of specific AGE adducts first, it then left ample space to fully and clearly describe the steps we took to characterize each AGE of interest (including NMR characterization for peptide **1^{MGH-1}** and peptide **1^{CEA}**, and chemical derivatization for peptide **1^{CEA}** and MGH-DH), and to link them mechanistically. **These new sections of text can be found in the revised manuscript.**

6. The text does not make clear whether relative intensity of mass peaks in the LC-MS experiments is strictly correlated with relative concentration of the particular glycation products in solution. The extent to which this was ascertained or calibrated should be clarified.

We agree that this is an important point to clarify. **We have added new text to the manuscript** that explains how our quantification approach (using % glycation) allows us to compare glycation levels across peptide variants that may have slight differences in ionization efficiency. Although we did not explicitly determine differences in ionization due to different AGEs, we would expect these differences to be fairly consistent between peptides, thus allowing us to compare AGE adduct distributions between discrete glycated peptides. **We have also added new text in the Supplementary Information** that clarifies how we quantified using a “molecular feature extractor” that tracks any and all charge states associated with a given ion, resulting in a more robust determination of abundance than integrating over a single peak area for a given charge state.

7. Several of the figures are exceedingly “busy”, and may try to provide too much information in too compact a space. It is therefore quite difficult for the reader to interpret some of these figures, which may be a particular problem for the broad readership of Nature Communications, who may not be previously familiar with glycochemistry. Particular glycation products are color-coded in the figures (e.g., Fig. 4B), but it is often necessary to cross-reference these color codes with previous figures in order to properly interpret the data. It’s fairly daunting, particularly when confronted with abbreviations in small font in the graph axes such as “hpd” (hours post dilution?).

We thank Reviewer 2 for pointing this out and apologize for these omissions in our prior version. In the revised submission, **we have revised all of the main text figures to make certain that each panel has its own legend and that any abbreviations are clearly stated in the figure display.** While we added significant new data to Fig. 4, we also edited the original data

Response to Reviewers

displayed in that figure, and feel that the revised version is more streamlined, clear and, as a result, significantly more impactful.

Reviewer #3

“This is a very well-written manuscript that sheds light on the site-specificity of non-enzymatic protein glycation events, a topic that is rarely considered or discussed. The figures are very well done and provide relatively clear results. While I believe this manuscript would be of interest to the Nature Communications readers, I think that a few major points need to be addressed. Providing an NMR of MG-H1 and MG-H3 is critical, as the authors claim that CEA can be generated from MG-H1 without providing this direct evidence (more on this below).”

We greatly appreciate this feedback from Reviewer 3. Our revised manuscript now includes an NMR and chemical derivatization of peptide **1^{CEA}**, in addition to the peptide **1^{MGH-1}** NMR that was included in the prior submission (Supplementary Fig. 16, Supplementary Fig. 17, and Supplementary Fig. 18). We also present new experiments that provide direct evidence that MGH-1 can be hydrolyzed to yield CEA (Figure 4 and Supplementary Fig. 19 and Supplementary Fig. 20). More detail about these changes can be found in the point-by-point response to Reviewer 3’s specific comments, below:

1. Why was the 3h timepoint chosen? Presumably shorter, or longer, incubations would result in a different profile of modifications. For example, the generation of CEA requires an intermediate (see DOI: 10.1021/ja301994d), so the full spectrum of modifications may be missed with this incubation period.

We completely agree with Reviewer 3 that time is an important factor that can influence AGE distributions, as we report in Fig. 4 and in several supplementary figures. **We have revised the main text to emphasize the reasons why we have selected the 3 h timepoint and have included a new Supplementary Fig. 8 along with additional discussion in multiple Supplementary Figure captions.** In particular, a newly added Supplementary Fig. 8 evaluates glycation at 3 and 24 h for “hit” peptide sequences identified from our library, along with Supplementary Fig. 14, which monitors peptide **1** glycation over a period of 4 weeks. In our past work,²⁵ we evaluated peptide glycation at multiple times, including 3 h and 24 h, and, in unpublished work, we’ve extensively monitored glycation at earlier time points (>3 h), intermediate time points (3-24 h), and for longer durations. This prior work has revealed that 3 h is an ideal time to check for preferential glycation, as we can see enough overall glycation that allows us to observe real differences in AGE distributions. Notably, we’ve found that these “early” differences are a more robust measure of the intrinsic glycation reactivity for a given peptide, particularly when the MGO concentration is equimolar with peptide. This is because, at later time points, the selectivity observed can appear to be eroded by the formation of AGEs that require multiple MGO molecules to react, or by rearrangements that share a common mechanistic intermediate. This is clearly observed in Fig. 4, where we compare time-dependent differences in glycation at high and low MGO concentrations, and is also apparent in Supplementary Fig. 3, Supplementary Fig. 8, Supplementary Fig. 14, and Supplementary Fig. 22.

2. As discussed and shown in Figure 3c, two distinct MG-H isomers can be observed, with MG-H1 being the predominate species. The mechanism shown in Figure 4e may not be correct: as shown by Galligan et al. (DOI: 10.1073/pnas.1802901115) and Wang et al. (see DOI: 10.1021/ja301994d), CEA results from the hydrolysis MG-H3, while MG-H1 remains (largely) stable. Did the authors confirm their findings via NMR? Further, if I am interpreting the figures correctly, it appears that the substitution of Phe at position +2 is sufficient to drive the formation of MG-H3, while other substitutions here do not. This would further push the formation of CEA, which is what is observed in 1e, 1f, and 1g, where the modifications are largely limited to CEA, rather than MG-H1. If this is true, this is perhaps equally as interesting as these findings now not only confer the relatively susceptibility of a given sequence to MG-H modification, but also the type of MG-H (and ultimately CEA) modification.

We appreciate these comments and agree that past work has explicitly stated that CEA can only form from MGH-3, even though our work demonstrates that MGH-1 can also produce CEA. In addition to

Response to Reviewers

the peptide $1^{\text{MGH-1}}$ NMR that was reported in our original submission, we have performed new experiments to confirm that the peptide 1 [M+72]^B adduct is peptide 1^{CEA} using both NMR and chemical derivatization. ***These new data appear in Supplementary Fig. 17 and Supplementary Fig. 18.***

As described in response to some of Reviewer 1's comments, we also purified MGH-1-modified peptide 1 (peptide $1^{\text{MGH-1}}$) and then incubated pure peptide $1^{\text{MGH-1}}$ in buffer for up to 48 hours and analyzed the reaction using LC-MS. We found that peptide $1^{\text{MGH-1}}$ is hydrolyzed directly to result in the formation of the peptide 1^{CEA} . Although those findings were reported in the supplementary materials from our prior submission, we performed a new and improved version of this experiment using different concentrations of purified peptide $1^{\text{MGH-1}}$, as shown in Fig. 4 and Supplementary Fig. 19. Using forcing conditions (pH = 12) to prepare larger quantities of peptide $1^{\text{MGH-1}}$ that were suitable for NMR studies, we also were able to identify a small peptide 1 [M+54] peak (most likely MGH-3) with a distinct retention time from the MGH-1 adduct. Although we could not obtain enough for NMR, we purified this adduct and found that the rate of its hydrolysis to CEA was greatly enhanced relative to the observed rate of hydrolysis for peptide $1^{\text{MGH-1}}$, as shown in Supplementary Fig. 20. This result provides further support to our conclusion that peptide $1^{\text{MGH-1}}$ is directly hydrolyzed to peptide 1^{CEA} . ***This new data can be found in Fig. 4, Supplementary Fig. 19 and Supplementary Fig. 20, along with additional text in the manuscript.***

We also found that the extent of hydrolysis of peptide $1^{\text{MGH-1}}$ was greater when lower concentrations of peptide were used. We suspect that this may reconcile past observations that MGH-1 is recalcitrant to hydrolysis, as prior studies with MGH-1 were performed at high (up to 6 mM³⁷) concentrations and/or using amino acid monomers that are unable to capture the influence of surrounding sidechains.³⁷⁻⁴⁰ Our studies used peptide substrates and spanned a range of low concentrations (1 μM – 1 mM, see Supplementary Fig. 19), which exhibit a clear increase in hydrolysis as peptide concentration decreases, and reflect near identical quantities of CEA observed during dilution experiments with peptide 1 . ***These new data can be found in main text Fig. 4 and Supplementary Fig. 19, along with additional discussion in the main text of the manuscript.*** Together, these data are most consistent with a model in which MGH-1 is directly converted to CEA, despite prior findings to the contrary by other groups.

We agree with Reviewer 3 that replacing Tyr in the +2 position with Phe leads to the formation of a second [M+54] adduct that is likely MGH-3. Past work has indicated that MGH-3 is the hydroimidazolone preferred by kinetics, whereas MGH-1 is more thermodynamically favorable. Thus, our interpretation of these results is that Tyr helps to drive formation of MGH-1 (rather than Phe driving MGH-3, as framed in the comment from Reviewer 3). It is certainly possible that the levels of CEA are somewhat enhanced during the dilution experiments for peptides **1e**, **1f**, and **1g**, though we did not comment on that in the revised manuscript, as it may be overly speculative. Either way, however, we strongly agree with Reviewer 3 that one of the most interesting conclusions of this work is that our "*findings now not only confer the relatively susceptibility of a given sequence to MG-H modification, but also the type of MG-H (and ultimately CEA) modification.*"

3. Can the authors speculate on the role of tertiary structure in conferring Arg susceptibility to MGO modification?

This is such an interesting question. Our prior work²⁵ suggested that the surrounding protein structure sculpts the exact glycation outcome (the AGE distribution) while the primary sequence governed the overall glycation susceptibility. Of course, we have also shown that primary sequence influences the AGE distribution. These conclusions are reconciled by simply considering glycation to be templated by the molecular environment—a combination of sequence and structure—surrounding a reactive site. In this sense, protein tertiary structure provides a well-defined microenvironment

Response to Reviewers

surrounding a reactive Arg, just as a short unstructured peptide provides a distinct, if looser, microenvironment that influences the glycation outcome. **We have added new text in the Conclusion section that shares our current thinking on this critical question.**

4. *The authors tend to focus on the type of modification being generated, which is obviously a major component of this manuscript; however, which peptide yielded the highest total amount of glycation. The graphs tend to show most peptides yield ~10-40% glycation, whereas some 6a-d are above 50%. I think this is an important consideration.*

It is true that the “peptide **6**” series, which contain mostly Gly and Tyr residues do lead to higher overall levels of glycation. However, most of this increased glycation is due to enhanced formation of an [M+72] and [M+144] adducts that most likely correlate with MGH-DH and THP, not the formation of MGH-1. The interplay between total glycation and formation of specific adducts is quite complicated, which can make it difficult to connect these observations meaningfully between peptides with very different sequences. We suspect this has to do with participation from neighboring side chains that can facilitate specific rearrangement steps, as we determined for the peptide **1** scaffold (see Fig. 5). **We have added new text to the Supplementary Fig 12 caption that explicitly comments on this phenomenon.**

5. *Confirmation of these findings on an intact protein would significantly increase the validity of these findings. Certainly, a complete proteomic survey of glycation sites is beyond the scope of this manuscript; however, can the authors identify sites on recombinant proteins with similar treatment conditions? A small survey of known glycated proteins would presumably yield sites with similar primary sequences as having the highest degree of modification. Further, this may yield interesting information on the susceptibility of Arg residues on a folded protein, rather than just focusing on the primary sequence.*

We completely agree with Reviewer 3 that it is important to demonstrate that our findings can translate to intact proteins, as our study seeks to provide useful information about how selective glycation is controlled in biological systems. To address this concern, we performed a new experiment that would allow us to determine if our conclusions remained applicable not only on a full-length, intact protein, but also one expressed in living mammalian cells. As described in our response to Reviewer 1, we expressed green fluorescent protein (GFP) variants that were C-terminally fused to the peptide **1** sequence (-LESRHYA, GFP-**1**), the peptide **3** sequence (-LDDREDA, GFP-**3**), or a negative control sequence inert to glycation (-LESAHYA, GFP-**1**^{Ala}) in HEK-293T cells. These C-terminal peptide sequences were connected to GFP through a linker sequence containing a tobacco etch virus (TEV) protease cleavage site. To monitor the glycation of GFP-**1**, GFP-**3**, and GFP-**1**^{Ala}, we treated cells with MGO following standard protocols.²⁹ After cells were harvested and lysed, we immunoprecipitated against GFP and then used TEV protease to liberate the C-terminal peptide (**1**, **3**, or **1**^{Ala}) fragment from GFP. Using LC-MS, we found that only GFP-**1** led to appreciable formation of an [M+54] adduct (likely MGH-1), as well as [M+72] adducts, whereas GFP-**3** yielded only vanishingly small quantities of [M+54] and GFP-**1**^{Ala} led to undetectable [M+54] levels. **These new results can be found in a newly added Fig. 6, with further data in new Supplementary Figs. 24 and 25, along with additional discussion in the manuscript.** Together, these data convincingly demonstrate that the chemical features we identified, which guide selective glycation in peptide substrates, are also relevant to the glycation of full-length protein targets expressed in living mammalian cells. We greatly appreciate this suggestion by Reviewer 3, as we feel that this experiment significantly strengthens our conclusions and enhances the impact of this manuscript.

6. *The evidence supporting the authors claim that “We also demonstrate, for the first time, that CEA is a hydrolysis product of MGH-1, and provide evidence that MGO concentrations influence glycation outcomes.” Is not entirely convincing. This has been shown to not be the case, again, see DOI: 10.1021/ja301994d. To demonstrate this point,*

Response to Reviewers

the authors will need to chemically synthesize MG-H1 and conduct a similar “hpd” experiment and quantify CEA. Based on *in vivo* measurements of MGO glycation profiles, it appears CEA is the most abundant modification (DOI: 10.1073/pnas.1802901115), with MG-H1 being second. Under these long-term conditions, should MG-H1 result in CEA, very little MG-H1 would be observed.

As previously described in our response to *comment #2* from Reviewer 3, we fully agree that past work has explicitly stated that CEA can only form from MGH-3, even though our work demonstrates that MGH-1 can also produce CEA. In addition to the peptide **1^{MGH-1}** NMR that was reported in our original submission, we have performed new experiments to confirm that the peptide **1 [M+72]^B** adduct is peptide **1^{CEA}** using both NMR and chemical derivatization. ***These new data appear in Supplementary Fig. 17 and Supplementary Fig. 18.*** We have also provided new experimental results that further bolster our conclusion that peptide **1^{MGH-1}** is directly hydrolyzed to peptide **1^{CEA}**. ***These new data can be found in Fig. 4, Supplementary Fig. 19 and Supplementary Fig. 20.*** Our data are most consistent with a model in which MGH-1 is directly converted to CEA, despite prior findings to the contrary by other groups.

We also agree with Reviewer 3 that, in light of our findings, it is very intriguing to consider the significance of the observation that “CEA is the most abundant modification (DOI: 10.1073/pnas.1802901115), with MG-H1 being second.” Our results suggest that MGH-1 is converted to CEA at low peptide (or protein) and MGO concentrations, thus the ratio of MGH-1 and CEA observed in biological samples could be a very important measure of exposure conditions, as they likely represent different points on a common mechanistic path. In our future studies, we are very interested to connect these chemical and mechanistic discoveries to the very complex and challenging world of glycation biology.

References:

- (1) Gao, Y.; Wang, Y. Site-Selective Modifications of Arginine Residues in Human Hemoglobin Induced by Methylglyoxal. *Biochemistry* **2006**, *45* (51), 15654–15660. <https://doi.org/10.1021/bi0614100>.
- (2) Ahmed, N.; Dobler, D.; Dean, M.; Thornalley, P. J. Peptide Mapping Identifies Hotspot Site of Modification in Human Serum Albumin by Methylglyoxal Involved in Ligand Binding and Esterase Activity. *J. Biol. Chem.* **2005**, *280* (7), 5724–5732. <https://doi.org/10.1074/jbc.M410973200>.
- (3) Brock, J. W. C.; Cotham, W. E.; Thorpe, S. R.; Baynes, J. W.; Ames, J. M. Detection and Identification of Arginine Modifications on Methylglyoxal-Modified Ribonuclease by Mass Spectrometric Analysis. *J. Mass Spectrom.* **2007**, *42* (1), 89–100. <https://doi.org/10.1002/jms.1144>.
- (4) Kimzey, M. J.; Kinsky, O. R.; Yassine, H. N.; Tsaprailis, G.; Stump, C. S.; Monks, T. J.; Lau, S. S. Site Specific Modification of the Human Plasma Proteome by Methylglyoxal. *Toxicol. Appl. Pharmacol.* **2015**, *289* (2), 155–162. <https://doi.org/10.1016/j.taap.2015.09.029>.
- (5) Oliveira, L. M.; Lages, A.; Gomes, R. A.; Neves, H.; Família, C.; Coelho, A. V.; Quintas, A. Insulin Glycation by Methylglyoxal Results in Native-like Aggregation and Inhibition of Fibril Formation. *BMC Biochemistry* **2011**, *12*, 41. <https://doi.org/10.1186/1471-2091-12-41>.
- (6) Cotham, W. E.; Metz, T. O.; Ferguson, P. L.; Brock, J. W. C.; Hinton, D. J. S.; Thorpe, S. R.; Baynes, J. W.; Ames, J. M. Proteomic Analysis of Arginine Adducts on Glyoxal-Modified Ribonuclease. *Mol Cell Proteomics* **2004**, *3* (12), 1145–1153. <https://doi.org/10.1074/mcp.M400002-MCP200>.
- (7) Gangadhariah, M. H.; Wang, B.; Linetsky, M.; Henning, C.; Spanneberg, R.; Glomb, M. A.; Nagaraj, R. H. Hydroimidazolone Modification of Human AA-Crystallin: Effect on the Chaperone Function and Protein Refolding Ability. *Biochim Biophys Acta* **2010**, *1802* (4), 432–441. <https://doi.org/10.1016/j.bbadis.2010.01.010>.

Response to Reviewers

- (8) Chen, Y.; Ahmed, N.; Thornalley, P. J. Peptide Mapping of Human Hemoglobin Modified Minimally by Methylglyoxal in Vitro. *Annals of the New York Academy of Sciences* **2005**, *1043* (1), 905–905. <https://doi.org/10.1196/annals.1333.119>.
- (9) Oya-Ito, T.; Naito, Y.; Takagi, T.; Handa, O.; Matsui, H.; Yamada, M.; Shima, K.; Yoshikawa, T. Heat-Shock Protein 27 (Hsp27) as a Target of Methylglyoxal in Gastrointestinal Cancer. *Biochimica et Biophysica Acta (BBA) - Molecular Basis of Disease* **2011**, *1812* (7), 769–781. <https://doi.org/10.1016/j.bbadis.2011.03.017>.
- (10) Godfrey, L.; Yamada-Fowler, N.; Smith, J.; Thornalley, P. J.; Rabbani, N. Arginine-Directed Glycation and Decreased HDL Plasma Concentration and Functionality. *Nutr Diabetes* **2014**, *4* (9), e134. <https://doi.org/10.1038/nuetd.2014.31>.
- (11) Ahmed, N.; Babaei-Jadidi, R.; Howell, S. K.; Beisswenger, P. J.; Thornalley, P. J. Degradation Products of Proteins Damaged by Glycation, Oxidation and Nitration in Clinical Type 1 Diabetes. *Diabetologia* **2005**, *48* (8), 1590–1603. <https://doi.org/10.1007/s00125-005-1810-7>.
- (12) Dobler, D.; Ahmed, N.; Song, L.; Eboigbodin, K. E.; Thornalley, P. J. Increased Dicarbonyl Metabolism in Endothelial Cells in Hyperglycemia Induces Anoikis and Impairs Angiogenesis by RGD and GFOGER Motif Modification. *Diabetes* **2006**, *55* (7), 1961–1969. <https://doi.org/10.2337/db05-1634>.
- (13) Scheckhuber, C. Q. Arg354 in the Catalytic Centre of Bovine Liver Catalase Is Protected from Methylglyoxal-Mediated Glycation. *BMC Res Notes* **2015**, *8*. <https://doi.org/10.1186/s13104-015-1793-5>.
- (14) Banerjee, S.; Maity, S.; Chakraborti, A. S. Methylglyoxal-Induced Modification Causes Aggregation of Myoglobin. *Spectrochimica Acta Part A: Molecular and Biomolecular Spectroscopy* **2016**, *155*, 1–10. <https://doi.org/10.1016/j.saa.2015.10.022>.
- (15) Oliveira, L. M. A.; Gomes, R. A.; Yang, D.; Dennison, S. R.; Família, C.; Lages, A.; Coelho, A. V.; Murphy, R. M.; Phoenix, D. A.; Quintas, A. Insights into the Molecular Mechanism of Protein Native-like Aggregation upon Glycation. *Biochimica et Biophysica Acta (BBA) - Proteins and Proteomics* **2013**, *1834* (6), 1010–1022. <https://doi.org/10.1016/j.bbapap.2012.12.001>.
- (16) Lima, M.; Moloney, C.; Ames, J. M. Ultra Performance Liquid Chromatography-Mass Spectrometric Determination of the Site Specificity of Modification of β -Casein by Glucose and Methylglyoxal. *Amino Acids* **2009**, *36* (3), 475–481. <https://doi.org/10.1007/s00726-008-0105-y>.
- (17) Gomes, R. A.; Oliveira, L. M. A.; Silva, M.; Ascenso, C.; Quintas, A.; Costa, G.; Coelho, A. V.; Sousa Silva, M.; Ferreira, A. E. N.; Ponces Freire, A.; Cordeiro, C. Protein Glycation in Vivo: Functional and Structural Effects on Yeast Enolase. *Biochem. J.* **2008**, *416* (3), 317–326. <https://doi.org/10.1042/BJ20080632>.
- (18) Augner, K.; Eichler, J.; Utz, W.; Pischetsrieder, M. Influence of Nonenzymatic Posttranslational Modifications on Constitution, Oligomerization and Receptor Binding of S100A12. *PLoS One* **2014**, *9* (11). <https://doi.org/10.1371/journal.pone.0113418>.
- (19) Watkins, N. G.; Thorpe, S. R.; Baynes, J. W. Glycation of Amino Groups in Protein. Studies on the Specificity of Modification of RNase by Glucose. *J. Biol. Chem.* **1985**, *260* (19), 10629–10636.
- (20) Silva, A. M. N.; Coimbra, J. T. S.; Castro, M. M.; Oliveira, Á.; Brás, N. F.; Fernandes, P. A.; Ramos, M. J.; Rangel, M. Determining the Glycation Site Specificity of Human Holo-Transferrin. *Journal of Inorganic Biochemistry* **2018**, *186*, 95–102. <https://doi.org/10.1016/j.jinorgbio.2018.05.016>.
- (21) Rabbani, N.; Godfrey, L.; Xue, M.; Shaheen, F.; Geoffrion, M.; Milne, R.; Thornalley, P. J. Glycation of LDL by Methylglyoxal Increases Arterial Atherogenicity. *Diabetes* **2011**, *60* (7), 1973–1980. <https://doi.org/10.2337/db11-0085>.
- (22) Terje Lund; Aud Svindland; Milaim Pepaj; Aase-Brith Jensen; Jens P Berg; Bente Kilhovd; Kristian F Hanssen. Fibrin(Ogen) May Be an Important Target for Methylglyoxal-Derived AGE Modification in Elastic Arteries of Humans. *Diabetes and Vascular Disease Research* **2011**, *8* (4), 284–294. <https://doi.org/10.1177/1479164111416831>.
- (23) Muranova, L. K.; Perfilov, M. M.; Serebryakova, M. V.; Gusev, N. B. Effect of Methylglyoxal Modification on the Structure and Properties of Human Small Heat Shock Protein HspB6 (Hsp20). *Cell Stress and Chaperones* **2016**, *21* (4), 617–629. <https://doi.org/10.1007/s12192-016-0686-4>.
- (24) Sakamoto, H.; Mashima, T.; Yamamoto, K.; Tsuruo, T. Modulation of Heat-Shock Protein 27 (Hsp27) Anti-Apoptotic Activity by Methylglyoxal Modification. *J. Biol. Chem.* **2002**, *277* (48), 45770–45775. <https://doi.org/10.1074/jbc.M207485200>.

Response to Reviewers

- (25) Sjoblom, N. M.; Kelsey, M. M. G.; Scheck, R. A. A Systematic Study of Selective Protein Glycation. *Angewandte Chemie International Edition* **2018**, *57* (49), 16077–16082. <https://doi.org/10.1002/anie.201810037>.
- (26) Zheng, Q.; Omans, N. D.; Leicher, R.; Osunsade, A.; Agustinus, A. S.; Finkin-Groner, E.; D'Ambrosio, H.; Liu, B.; Chandarlapaty, S.; Liu, S.; David, Y. Reversible Histone Glycation Is Associated with Disease-Related Changes in Chromatin Architecture. *Nature Communications* **2019**, *10* (1), 1289. <https://doi.org/10.1038/s41467-019-09192-z>.
- (27) Zheng, Q.; Osunsade, A.; David, Y. Protein Arginine Deiminase 4 Antagonizes Methylglyoxal-Induced Histone Glycation. *Nature Communications* **2020**, *11* (1), 3241. <https://doi.org/10.1038/s41467-020-17066-y>.
- (28) Bollong, M. J.; Lee, G.; Coukos, J. S.; Yun, H.; Zambaldo, C.; Chang, J. W.; Chin, E. N.; Ahmad, I.; Chatterjee, A. K.; Lairson, L. L.; Schultz, P. G.; Moellering, R. E. A Metabolite-Derived Protein Modification Integrates Glycolysis with KEAP1–NRF2 Signalling. *Nature* **2018**, *562* (7728), 600. <https://doi.org/10.1038/s41586-018-0622-0>.
- (29) Uchiki, T.; Weikel, K. A.; Jiao, W.; Shang, F.; Caceres, A.; Pawlak, D.; Handa, J. T.; Brownlee, M.; Nagaraj, R.; Taylor, A. Glycation-Altered Proteolysis as a Pathobiologic Mechanism That Links Dietary Glycemic Index, Aging, and Age-Related Disease (in Non Diabetics). *Aging Cell* **2012**, *11* (1), 1–13. <https://doi.org/10.1111/j.1474-9726.2011.00752.x>.
- (30) Venkatraman, J.; Aggarwal, K.; Balaram, P. Helical Peptide Models for Protein Glycation: Proximity Effects in Catalysis of the Amadori Rearrangement. *Chemistry & Biology* **2001**, *8* (7), 611–625. [https://doi.org/10.1016/S1074-5521\(01\)00036-9](https://doi.org/10.1016/S1074-5521(01)00036-9).
- (31) Iberg, N.; Flückiger, R. Nonenzymatic Glycosylation of Albumin in Vivo. Identification of Multiple Glycosylated Sites. *J. Biol. Chem.* **1986**, *261* (29), 13542–13545.
- (32) Shapiro, R.; McManus, M. J.; Zalut, C.; Bunn, H. F. Sites of Nonenzymatic Glycosylation of Human Hemoglobin A. *J. Biol. Chem.* **1980**, *255* (7), 3120–3127.
- (33) Shilton, B. H.; Campbell, R. L.; Walton, D. J. Site Specificity of Glycation of Horse Liver Alcohol Dehydrogenase in Vitro. *European Journal of Biochemistry* **1993**, *215* (3), 567–572. <https://doi.org/10.1111/j.1432-1033.1993.tb18067.x>.
- (34) Cai, W.; Gao, Q.-D.; Zhu, L.; Peppas, M.; He, C.; Vlassara, H. Oxidative Stress-Inducing Carbonyl Compounds from Common Foods: Novel Mediators of Cellular Dysfunction. *Mol Med* **2002**, *8* (7), 337–346.
- (35) Wang, T.; Streeter, M. D.; Spiegel, D. A. Generation and Characterization of Antibodies against Arginine-Derived Advanced Glycation Endproducts. *Bioorganic & Medicinal Chemistry Letters* **2015**, *25* (21), 4881–4886. <https://doi.org/10.1016/j.bmcl.2015.06.013>.
- (36) Lam, K. S.; Lebl, M.; Krchňák, V. The “One-Bead-One-Compound” Combinatorial Library Method. *Chem. Rev.* **1997**, *97* (2), 411–448. <https://doi.org/10.1021/cr9600114>.
- (37) Klöpfer, A.; Spanneberg, R.; Glomb, M. A. Formation of Arginine Modifications in a Model System of N α -Tert-Butoxycarbonyl (Boc)-Arginine with Methylglyoxal. *J. Agric. Food Chem.* **2011**, *59* (1), 394–401. <https://doi.org/10.1021/jf103116c>.
- (38) Galligan, J. J.; Wepy, J. A.; Streeter, M. D.; Kingsley, P. J.; Mitchener, M. M.; Wauchope, O. R.; Beavers, W. N.; Rose, K. L.; Wang, T.; Spiegel, D. A.; Marnett, L. J. Methylglyoxal-Derived Posttranslational Arginine Modifications Are Abundant Histone Marks. *PNAS* **2018**, *115* (37), 9228–9233. <https://doi.org/10.1073/pnas.1802901115>.
- (39) Wang, T.; Kartika, R.; Spiegel, D. A. Exploring Post-Translational Arginine Modification Using Chemically Synthesized Methylglyoxal Hydroimidazolones. *J. Am. Chem. Soc.* **2012**, *134* (21), 8958–8967. <https://doi.org/10.1021/ja301994d>.
- (40) Ahmed, N.; Thornalley, P. J. Chromatographic Assay of Glycation Adducts in Human Serum Albumin Glycated in Vitro by Derivatization with 6-Aminoquinolyl-N-Hydroxysuccinimidyl-Carbamate and Intrinsic Fluorescence. *Biochemical Journal* **2002**, *364* (1), 15–24. <https://doi.org/10.1042/bj3640015>.

Reviewers' Comments:

Reviewer #1:

Remarks to the Author:

The authors response to my comments are not satisfactory.

I appreciate that Author's accepted that MG used in this study is impure. However, their suggestion in response to comment 1 that impurity may reflect the physiological system, lack scientific integrity. Clearly if you are designing an experiment to study the effect of only MG you need to add to your test experiments only MG. New experiment to address mine and the reviewer 3's concerns about the validity of the study is based on GFP linked peptide in the HEK 293T cells. GFP study is totally bias towards their intended outcome by designing an experiment with extreme super physiological increase in copy number of GFP peptides compare to all other protein motifs in HEK293T cell back ground.

I would insist that they must provide data on intact protein to confirm the validity of the study finding, without the suggested experiment, the study is barely a reaction of chemically synthesized short peptides with Impure MG.

Reviewer #2:

Remarks to the Author:

The authors have revised their manuscript extensively in order to respond to all of the reviewers' comments through additional experiments, analysis and commentary. Although there are clear differences between some of the conclusions drawn in this report and previous studies on protein glycation, the authors provide appropriate justification for their results and conclusions.

I have some residual concerns that the study relies excessively on the proper function of an individual commercial anti alpha-MGH-1 antibody, and the sequence of the peptides identified did not match that used to obtain this antibody (in this regard it is notable that an alternative experimental approach evaluating impaired proteolysis by trypsin yielded results that were not consistent with the results presented here). In addition, I am somewhat surprised that only 75 beads were detected as binding to this antibody out of 120,000 beads screened.

Nevertheless, I am overall very satisfied with the revised manuscript and believe it is suitable for publication in its current form.

I also note that the authors response to the reviewers' comments is among the more professional, thoughtful and thorough responses that I have evaluated.

Reviewer #3:

Remarks to the Author:

This manuscript is markedly improved following revision. While I believe the authors have addressed most of my concerns, their approach to addressing the relevance of these findings in cells is a bit confusing. As per the original revision, "Confirmation of these findings on an intact protein would significantly increase the validity of these findings. Certainly, a complete proteomic survey of glycation sites is beyond the scope of this manuscript; however, can the authors identify sites on recombinant proteins with similar treatment conditions? A small survey of known glycated proteins would presumably yield sites with similar primary sequences as having the highest degree of modification. Further, this may yield interesting information on the susceptibility of Arg residues on a folded protein, rather than just focusing on the primary sequence."

The authors have seemingly attempted to address this using the GFP approach; however, by fusing a known glycated peptide to this approach they are inherently biasing their results. It is unclear why a more traditional approach, using an intact, fully folded protein was not used with a known glycated protein (e.g. histones, albumin). Sadly, I don't believe that the experiment completed addresses the question(s) at hand.

Response to Reviewers

Reviewer #3

1. *"This manuscript is markedly improved following revision. While I believe the authors have addressed most of my concerns, their approach to addressing the relevance of these findings in cells is a bit confusing."*

We greatly appreciated Reviewer 3's original comments and suggestions during the first round of review and, in response, we performed a significant number of new experiments to address them; we are therefore pleased that Reviewer 3 agrees that our revised manuscript is markedly improved. We seek to clarify our approach on a point-by-point basis below, and we have also made substantial revisions to the manuscript text to address the remaining feedback from Reviewer 3:

2. *"As per the original revision, "Confirmation of these findings on an intact protein would significantly increase the validity of these findings. Certainly, a complete proteomic survey of glycation sites is beyond the scope of this manuscript; however, can the authors identify sites on recombinant proteins with similar treatment conditions? A small survey of known glycated proteins would presumably yield sites with similar primary sequences as having the highest degree of modification. Further, this may yield interesting information on the susceptibility of Arg residues on a folded protein, rather than just focusing on the primary sequence."*

When we received Reviewer 3's original comments, we agreed that *"confirmation of these findings on an intact protein would significantly increase the validity of these findings."* We understood this statement to reflect Reviewer 3 wondering, quite fairly, if the results suggested by our *in vitro* work (Figs. 2-5) would remain relevant in a biological context. After careful consideration of Reviewer 3's original comments, we designed the *"GFP approach"*. As described in the response to comment #3, this approach allowed us to draw the strongest conclusions about the specific *in vitro* results identified in this manuscript, while complementing and not retreading our previously published work (see also below).¹ Accordingly, the experiment we performed (Fig. 6) does indeed demonstrate that our findings about the features that control selective glycation for peptides *in vitro* remain applicable on full-length intact proteins in living cells.

Reviewer 3 suggested that a *"small survey of known glycated proteins would presumably yield sites with similar primary sequences as having the highest degree of modification."* In fact, **we have already performed that study, which was published in *Angewandte Chemie* in 2018 and is cited extensively in the current manuscript (ref. 38).**¹ Reviewer 3 is correct that our past study, in which we evaluated glycation under identical conditions for a panel of nearly 20 intact, purified proteins, provided *"interesting information on the susceptibility of Arg residues on a folded protein."* Indeed, we learned that selective glycation is not correlated with a lowered Arg pK_a or with increased solvent exposure, as suggested in past work.²⁻⁶ Instead, our study revealed that primary sequence is a major driver for glycation, broadly dictating the overall susceptibility for a site to become glycated as well as influencing the specific AGE distribution at that site. Notably, Reviewer 3's prediction is incorrect: this prior study did not reveal *"similar primary sequences as having the highest degree of modification."* Instead, our 2018 report suggested critical trends based on nearby residues (not necessarily primary sequence). We then performed a series of experiments to test these hypotheses using short, unstructured peptides and found that glycation can indeed be governed by primary sequence. Thus, we concluded that clustered negative charge is detrimental for glycation, whereas a combination of polar residues and Tyr appear to be beneficial for glycation. These trends represent broad chemical guidelines that remain true irrespective of the individual AGEs that form. In the current study, we have substantiated and expanded these original findings using an entirely different *in vitro* approach. Further, in the current study, we have progressed the ideas to discern the specific ways in which primary sequence influences the formation of discrete AGEs, like MGH-1. Our results about the

Response to Reviewers

features that promote MGH-1, which we have shown to remain true in cells, provide new chemical detail regarding the formation of one of the most abundant cellular AGEs.

Further, we agree with Reviewer 3 that “*a complete proteomic survey of glycation sites is beyond the scope of this manuscript.*” Moreover, from our experience working in the glycation field, such an approach is also unlikely to reveal the features that bias certain types of glycation outcomes. This is because proteomic studies are well-suited for identifying AGE-modified cellular proteins,^{7–10} but struggle with the heterogeneity intrinsic to cellular glycation. For instance, cells contain many biologically-relevant aldehydes,^{11–18} each of which can influence the preferred sites of glycation and form many distinct and/or isomeric AGEs.^{2,19,20} Cell-based proteomic experiments can also introduce artifacts that arise from differential protein expression levels in cells. Thus, cataloging cellular glycation events is unlikely to reveal the underlying chemical features that govern preferential glycation, which is the scientific question we sought to address in this study. Taking an *in vitro* approach like the one used in this study enables us to evaluate the fundamental question about the underlying chemical features that promote glycation, rather than simply enumerating cellular glycation events.

Past proteomics studies,^{7,21–23} and our own prior work¹ have clearly demonstrated that there is no ‘consensus sequence’ that controls selective glycation. Our work has provided a rationale for this finding: glycation is influenced not only by primary sequence, but also by the nearby polar and ionizable residues in proximity to a reactive Arg. The combination of surrounding sequence and structure provide a chemical microenvironment that facilitates critical rate-determining steps that lead to formation of stable AGEs like MGH-1. Because of this, we knew at the outset that **experimental approaches that rely on unbiased cell-based proteomics or targeted studies of overexpressed, known glycated proteins (as suggested by Reviewer 3) would be unable to report on our current findings** that focus on the fundamental question of how selective glycation is controlled. Thus, we turned to an alternative approach that allowed us to evaluate explicitly if the sequence effects we identified *in vitro* remain relevant for a full-length, folded protein in a cellular environment, as described in further detail below. However, upon reading Reviewer 3’s comments after our initial revision, we realize that we did not provide enough context to explain our experimental choice. In particular, we did not adequately acknowledge our own relevant background work, or past proteomics studies by others, that provides critical rationale for why a seemingly more straightforward experiment (like “*overexpression of a histone*”) would be unable to provide a meaningful result in this context. We deeply apologize for this omission in our earlier submission, and to address this problem ***we have added significant new text in our newly revised manuscript (highlighted in blue), which provides the relevant background and context for our approach.*** We appreciate this feedback from Reviewer 3, and we believe that this substantial new addition to the text will add significant clarity for future readers of this work.

3. “*The authors have seemingly attempted to address this using the GFP approach; however, by fusing a known glycated peptide to this approach they are inherently biasing their results.*”

As described above, after careful consideration of Reviewer 3’s original comments, we designed the “*GFP approach*”, in which “hit” (peptide **1**) and “non-hit” (peptide **3**) peptides identified and validated from our library were fused to GFP, expressed in mammalian cells as full-length protein constructs, and evaluated for their susceptibility to become glycated. In our view, this experiment is a beautifully direct way to evaluate the transferability and applicability of our *in vitro* findings to a protein and cellular context. This approach allowed us to draw the strongest conclusions about the specific *in vitro* results identified in this manuscript, while complementing and not retreading our previously published work (see also above).¹ Accordingly, the experiment we performed (Fig. 6) does indeed

Response to Reviewers

demonstrate that our findings about the features that control selective glycation for peptides *in vitro* remain applicable on full-length intact proteins in living cells.

We are uncertain as to why Reviewer 3 feels that the GFP experiment we performed does not effectively address the aforementioned concern that “*confirmation of these findings on an intact protein would significantly increase the validity of these findings.*” The results of our extensive *in vitro* studies confirmed that peptide **1** (LESRHYA) is glycosylated to a greater extent than peptide **3** (LDDREDA) and also forms significantly greater levels of an [M+54] adduct corresponding to MGH-1. To evaluate if such findings are meaningful for understanding how glycation is controlled on protein substrates and in cells, we fused these peptides to full-length GFP (GFP-**1** and GFP-**3**, respectively). We then expressed GFP-**1** and GFP-**3** in mammalian cells that were treated with MGO. Afterwards, we tested the hypothesis that GFP-**1** would exhibit higher levels of glycation and more [M+54] than GFP-**3** using mass spectrometry. This experiment **directly addresses the question of whether or not the rules that control selective glycation of peptides *in vitro* also remain relevant for controlling the glycation of proteins expressed in cells.** This experiment clearly demonstrates that the answer to this question is “yes.” We are delighted that our work has provided the necessary groundwork not only to further investigate MGH-1 glycation, but also to develop new tools to control selective glycation in the future, enabling its study as a functional PTM.

Reviewer 3 states that “*by fusing a known glycosylated peptide to this approach they are inherently biasing their results.*” We are extremely puzzled by this statement. To us, it is unclear why Reviewer 3 refers to the peptide **1** or peptide **3** GFP fusions as “*known glycosylated peptides*” because both of those sequences were only just identified in the current work that is now under consideration. Reviewer 3’s original critique was very reasonable, asking us to design and perform a crucial validation experiment to ask if that the brand-new results we generated *in vitro* do indeed remain applicable in a cellular context. Given that the sequences LESRHYA and LDDREDA are not derived from a specific mammalian protein, the only way to perform a true test of our *in vitro* results was to fuse these sequences to a carrier protein. We were hopeful, but not certain, that this experiment would demonstrate that the GFP-**1** fusion would lead to greater levels of glycation and [M+54] adducts than GFP-**3**. That is, of course, why we felt it was important to do, and we are grateful to Reviewer 3 for the suggestion. As described above, to address this feedback from Reviewer 3, **we have added substantial new text to the manuscript in this second round of revision to better describe the context and motivation for the experiment we performed.** Ultimately, we are extremely pleased that this experiment confirmed that the sequence effects we identified *in vitro* remain relevant for a full-length protein fusion expressed in cells, establishing that our *in vitro* approach is able to provide useful information about how selective glycation is governed in a cellular environment.

4. “*It is unclear why a more traditional approach, using an intact, fully folded protein was not used with a known glycosylated protein (e.g. histones, albumin).*”

As described above, we have already performed this study, which was published in *Angewandte Chemie* in 2018 and is cited extensively in the current manuscript (ref. 38).¹ In that study, we evaluated glycation under identical conditions for a small panel of intact proteins, including known glycosylated proteins like albumins, crystallins, and hemoglobin, to name a few. To address this comment, **we have added substantial new text to the manuscript in this second round of revision to better describe our relevant prior work.**

5. “*Sadly, I don’t believe that the experiment completed addresses the question(s) at hand.*”

From the most recent Reviewer comments, it seems that Reviewer 3 may have been expecting an experiment reminiscent of our previously published work and/or a targeted proteomics study of

Response to Reviewers

overexpressed native proteins that are known to be glycosylated. In particular, we agree that experiments focused on the glycosylation of native proteins would, potentially, be quite interesting. However, based on our current understanding of selective glycosylation, those experiments unavoidably conflate multiple, distinct questions, including:

- i. Are the trends seen *in vitro* relevant to glycosylation of intact proteins by MGO in mammalian cells?
- ii. Are the trends seen *in vitro* affected by secondary structure, tertiary structure, T_m , or other properties of folded proteins?
- iii. Can the trends seen *in vitro* be used to predict glycosylation at sites on fully folded native proteins, in a proteome-wide fashion?

The experiment suggested by Reviewer 3 (“*overexpression of a histone (for example)*”) would not provide meaningful results because it conflates all three of these questions, sacrificing the nuance required for each answer. Moreover, given the complex nature of glycosylation, it is simply not possible to address all three questions in a single experiment with just one or two native proteins. By contrast, **the experiment that we performed addresses only the one most critical question, the first: Are the trends seen *in vitro* relevant to glycosylation of intact proteins by MGO in mammalian cells?** In particular, the experiment we chose to perform enabled us to validate the most novel findings reported in our current study while advancing our collective understanding of how selective glycosylation is controlled, not simply repeating our past work (see also the response to comments #2-#4). As the Reviewers note, answering the second two questions will require extensive additional work that is hardly suitable as “add-on” experiments to the current manuscript. And while our lab is certainly eager to answer those second two questions, just as the Reviewers are, and this work is already underway in our lab, it represents years of further work that is outside of the scope of the current manuscript. Thus, answering the first question conclusively was the most important and relevant for this current work, and that is why the approach using GFP fusions is ideally suited for our current study.

We regret that we overlooked the importance of providing the critical background—from our own work and the work of others—that explains why we did not choose other approaches, including a targeted study of overexpressed native proteins that are known to be glycosylated. As described above, to address this feedback from Reviewer 3, **we have added substantial new text to the manuscript to better describe the context and motivation for the experiment we performed.** Indeed, the GFP experiment in this work definitively demonstrates that we have uncovered molecular features that control selective glycosylation, not only for peptides *in vitro*, but also for full-length proteins expressed in cells. In our opinion, while Reviewer 3 may not have been expecting the experiment we performed, we believe it does an excellent job of addressing the one most important question at hand: whether the results suggested by our *in vitro* work (Figs. 2-5) remain relevant in a biological context. The experiment we performed (Fig. 6) does indeed demonstrate that our findings about the features that control selective glycosylation for peptides *in vitro* remain applicable on full-length intact proteins in living cells.

Reviewer #2

1. *“The authors have revised their manuscript extensively in order to respond to all of the reviewers' comments through additional experiments, analysis and commentary. Although there are clear differences between some of the conclusions drawn in this report and previous studies on protein glycosylation, the authors provide appropriate justification for their results and conclusions.”*

Response to Reviewers

We greatly appreciated Reviewer 2 comments and suggestions during the first round of revision and performed a significant number of new experiments to address them; we are therefore pleased that Reviewer 2 agrees that our manuscript has been improved through our extensive revision, including significant new experiments, along with additional analysis and commentary. We further appreciate Reviewer 2's acknowledgement that, although there are clear differences between our findings and past studies, our results and conclusions are appropriately justified.

- I have some residual concerns that the study relies excessively on the proper function of an individual commercial anti alpha-MGH-1 antibody, and the sequence of the peptides identified did not match that used to obtain this antibody (in this regard it is notable that an alternative experimental approach evaluating impaired proteolysis by trypsin yielded results that were not consistent with the results presented here). In addition, I am somewhat surprised that only 75 beads were detected as binding to this antibody out of 120,000 beads screened.*

This is an interesting point. First, we extensively optimized the screening conditions to keep the number of potential “hits” fairly low. This is because, from our past work on glycation, we knew that the effects can be subtle and hard to disentangle. By keeping the stringency high during screening, there were fewer hits selected, making it a bit easier to tease out the features that were having a large influence on glycation in our follow up studies. Regarding the alternative screening approach using trypsin, it is true that the consensus motif that was obtained is different from the one obtained when using α -MGH-1 antibodies. However, we believe that these results are consistent with a model in which the “trypsin” approach selects for any modified glycation adduct, which could include more than ten known chemical structures that can form just from the reaction between MGO and Arg.²⁴⁻²⁶ It is most likely that the trypsin approach screens for the first steps of the glycation reaction that occur rapidly, such as the formation of MGH-DH (Fig. 4f), whereas the α -MGH-1 screening approach provides information about features that promote the (presumably rate-determining) elimination step that converts MGH-DH to MGH-1. We had additional discussion about this in earlier drafts of the manuscript, but it was removed due to space constraints, as we felt it was not as essential to our conclusions as some of our other results. However, we greatly appreciate the curiosity and are happy to be able to share our thinking on this important topic.

- “Nevertheless, I am overall very satisfied with the revised manuscript and believe it is suitable for publication in its current form. I also note that the authors response to the reviewers' comments is among the more professional, thoughtful and thorough responses that I have evaluated.”*

Thank you, these comments are greatly appreciated.

Reviewer #1

- “The authors response to my comments are not satisfactory. I appreciate that Author's accepted that MG used in this study is impure. However, their suggestion in response to comment 1 that impurity may reflect the physiological system, lack scientific integrity. Clearly if you are designing an experiment to study the effect of only MG you need to add to your test experiments only MG.”*

To clarify, we do not agree with Reviewer 1's original assertion that it is a problem to use commercial MGO. We note that Reviewer 1 has not provided any references suggesting that MGO we are using is impure. We are simply stating that, from a chemical perspective, any solution (even a recently distilled solution) of an enolizable 1,2-dicarbonyl like MGO will be in a dynamic equilibrium with itself. Thus, it is meaningless in the context of the experiments described in this work to insist that MGO could ever be “100% pure”. We previously noted that we have not noticed any major differences

Response to Reviewers

that result from different bottles of commercial MGO or that are introduced over extended storage using the manufacturer recommended conditions. We also performed a number of studies using purified AGE-modified peptides, which support our initial findings, further eliminating any concerns about the source of MGO interfering with our results. We also point out that numerous recent publications in this journal^{27,28} and other top journals²⁹ have performed well-controlled studies using the same commercial source of MGO without further purification, and we have published our prior work in *Angewandte Chemie*¹ using these same materials. The reality that MGO cross-reacts with itself and other cellular molecules makes the experiment we performed in live cells (Fig. 6) even more critical. This experiment, which we performed during the first revision of our manuscript, directly confirms that the features that control selective glycation of peptides *in vitro* also remain relevant for full-length, folded proteins expressed in cells. From this we conclude that **the commercial source of MGO that we used for our *in vitro* studies is appropriate for generating useful information about how selective glycation is controlled, both *in vitro* and in a cellular environment.**

We are confused as to why Reviewer 1 thinks that we lack scientific integrity by pointing out that it is not scientifically accurate to suggest that MGO can be “pure” when doing experiments in complex environments such as the cell. Mammalian cells are estimated to contain 300 mM total metabolites and 3 mM total protein, while typical MGO concentrations are thought to be 1-5 μM .³⁰⁻³² Thus, MGO is just one of many biologically-relevant aldehydes (not to mention metabolites in general) that each has the potential to cross-react and/or form side products in addition to the known AGEs.

2. *“New experiment to address mine and the reviewer 3’s concerns about the validity of the study is based on GFP linked peptide in the HEK 293T cells. GFP study is totally bias towards their intended outcome by designing an experiment with extreme super physiological increase in copy number of GFP peptides compare to all other protein motifs in HEK293T cell back ground.”*

It seems that Reviewer 1 is suggesting that overexpression of a protein in a mammalian cell an inappropriate experimental approach. Overexpression of proteins in mammalian cells is widely used across many disciplines, including cell biology and chemical biology. Even just limiting ourselves to the glycation literature, we found a number of recent studies in top journals that used a similar approach.^{27,28,33-36}

Additionally, we did not claim that the glycation of GFP-1 or GFP-3 is physiologically relevant in a native sense, as neither sequence occurs naturally in the mammalian proteome. We also were not comparing the glycation of these constructs to native proteins in the HEK-293T cell proteome, which is the only scenario in which the GFP copy number might bias the results. Instead, we used this experiment to demonstrate that our findings about the features that control selective glycation remain true on a full-length, folded protein target expressed in mammalian cells (for experimental details, please see the response to *comment #3* from Reviewer 3). Because we were comparing the glycation of GFP-1 to GFP-3 (which were expressed comparably), **the design of the experiment is completely appropriate.** Even though overexpression introduced the GFP fusions at high levels, we can still conclude that GFP-1 is glycated to a greater extent than GFP-3 and also forms more of an [M+54] product. As a result, this experiment is quite effective at demonstrating that the sequence features we identified that control selective glycation for peptides *in vitro* remain applicable on full-length intact proteins in living cells.

3. *“I would insist that they must provide data on intact protein to confirm the validity of the study finding, without the suggested experiment, the study is barely a reaction of chemically synthesized short peptides with Impure MG.”*

Response to Reviewers

As we described above, we have already performed this study, which was published in *Angewandte Chemie* in 2018 and is cited extensively in the current manuscript (ref. 38).¹ However, we have also already included a new experiment that used GFP fusions to confirm the validity of our study findings, not only on full-length intact proteins, but also inside of a mammalian cell. As we explained above (see *comment #5* from Reviewer 3), this experiment does not address the effects of secondary/tertiary structure on glycation, which is a complex question that we are currently pursuing in ongoing work in our lab. However, these efforts represent years of further work that is outside of the scope of the current manuscript and cannot be addressed with one or two more experiments with just a single intact protein. The GFP experiment we already provided does address the most relevant question: if the features that control selective glycation for peptides *in vitro* remain applicable on full-length intact proteins in living cells. This experiment we performed clearly demonstrates that the answer to this question is “yes.” As described above, **we have added substantial new text to the manuscript in this second round of revision to better describe the context and motivation for the GFP experiment that we performed.**

To call our study “barely a reaction of chemically synthesized short peptides with Impure MG” seems, at best, incomplete. It is true that the bulk of this study was performed using synthetic peptides and a commercial source of MGO. However, this unique and innovative approach enabled us to significantly advance our collective understanding of glycation: Our study is the first to reveal specific mechanistic steps connecting multiple AGEs, including MGH-DH, MGH-1, and CEA. We also demonstrate, for the first time, that CEA is a hydrolysis product of MGH-1 and provide evidence that both peptide and MGO concentrations influence glycation outcomes. Our study has found that tyrosine plays an active mechanistic role that facilitates MGH-1 formation, and provides clear evidence that glycation outcomes can be influenced through long- or medium-range cooperative interactions. Our work further demonstrates that these chemical features also predictably template selective glycation on full-length protein targets expressed in mammalian cells. With all of these significant, novel, and long-sought findings, it seems that the approach we have used is quite justified.

References:

- (1) Sjoblom, N. M.; Kelsey, M. M. G.; Scheck, R. A. A Systematic Study of Selective Protein Glycation. *Angewandte Chemie International Edition* **2018**, *57* (49), 16077–16082. <https://doi.org/10.1002/anie.201810037>.
- (2) Ahmed, N.; Dobler, D.; Dean, M.; Thornalley, P. J. Peptide Mapping Identifies Hotspot Site of Modification in Human Serum Albumin by Methylglyoxal Involved in Ligand Binding and Esterase Activity. *J. Biol. Chem.* **2005**, *280* (7), 5724–5732. <https://doi.org/10.1074/jbc.M410973200>.
- (3) Venkatraman, J.; Aggarwal, K.; Balaram, P. Helical Peptide Models for Protein Glycation: Proximity Effects in Catalysis of the Amadori Rearrangement. *Chemistry & Biology* **2001**, *8* (7), 611–625. [https://doi.org/10.1016/S1074-5521\(01\)00036-9](https://doi.org/10.1016/S1074-5521(01)00036-9).
- (4) Iberg, N.; Flückiger, R. Nonenzymatic Glycosylation of Albumin in Vivo. Identification of Multiple Glycosylated Sites. *J. Biol. Chem.* **1986**, *261* (29), 13542–13545.
- (5) Shapiro, R.; McManus, M. J.; Zalut, C.; Bunn, H. F. Sites of Nonenzymatic Glycosylation of Human Hemoglobin A. *J. Biol. Chem.* **1980**, *255* (7), 3120–3127.
- (6) Shilton, B. H.; Campbell, R. L.; Walton, D. J. Site Specificity of Glycation of Horse Liver Alcohol Dehydrogenase in Vitro. *European Journal of Biochemistry* **1993**, *215* (3), 567–572. <https://doi.org/10.1111/j.1432-1033.1993.tb18067.x>.
- (7) Bilova, T.; Paudel, G.; Shilyaev, N.; Schmidt, R.; Brauch, D.; Tarakhovskaya, E.; Milrud, S.; Smolikova, G.; Tissier, A.; Vogt, T.; Sinz, A.; Brandt, W.; Birkemeyer, C.; Wessjohann, L. A.; Frolov, A. Global Proteomic Analysis of Advanced Glycation End Products in the Arabidopsis Proteome Provides Evidence for Age-Related Glycation Hot Spots. *J. Biol. Chem.* **2017**, *292* (38), 15758–15776. <https://doi.org/10.1074/jbc.M117.794537>.

Response to Reviewers

- (8) Bilova, T.; Lukasheva, E.; Brauch, D.; Greifenhagen, U.; Paudel, G.; Tarakhovskaya, E.; Frolova, N.; Mittasch, J.; Balcke, G. U.; Tissier, A.; Osmolovskaya, N.; Vogt, T.; Wessjohann, L. A.; Birkemeyer, C.; Milkowski, C.; Frolov, A. A Snapshot of the Plant Glycated Proteome: Structural, Functional and Mechanistic Aspects. *J. Biol. Chem.* **2016**, jbc.M115.678581. <https://doi.org/10.1074/jbc.M115.678581>.
- (9) Zhang, Q.; Monroe, M. E.; Schepmoes, A. A.; Clauss, T. R. W.; Gritsenko, M. A.; Meng, D.; Petyuk, V. A.; Smith, R. D.; Metz, T. O. Comprehensive Identification of Glycated Peptides and Their Glycation Motifs in Plasma and Erythrocytes of Control and Diabetic Subjects. *J. Proteome Res.* **2011**, *10* (7), 3076–3088. <https://doi.org/10.1021/pr200040j>.
- (10) Gomes, R. A.; Miranda, H. V.; Silva, M. S.; Graça, G.; Coelho, A. V.; Ferreira, A. E.; Cordeiro, C.; Freire, A. P. Yeast Protein Glycation in Vivo by Methylglyoxal. *FEBS Journal* **2006**, *273* (23), 5273–5287. <https://doi.org/10.1111/j.1742-4658.2006.05520.x>.
- (11) Thornalley, P. J. Dicarbonyl Intermediates in the Maillard Reaction. *Annals of the New York Academy of Sciences* **2005**, *1043* (1), 111–117. <https://doi.org/10.1196/annals.1333.014>.
- (12) Thornalley, P. J.; Battah, S.; Ahmed, N.; Karachalias, N.; Agalou, S.; Babaei-Jadidi, R.; Dawnay, A. Quantitative Screening of Advanced Glycation Endproducts in Cellular and Extracellular Proteins by Tandem Mass Spectrometry. *Biochemical Journal* **2003**, *375* (3), 581–592. <https://doi.org/10.1042/bj20030763>.
- (13) Rabbani, N.; Thornalley, P. J. Methylglyoxal, Glyoxalase 1 and the Dicarbonyl Proteome. *Amino Acids* **2012**, *42* (4), 1133–1142. <https://doi.org/10.1007/s00726-010-0783-0>.
- (14) Oya, T.; Hattori, N.; Mizuno, Y.; Miyata, S.; Maeda, S.; Osawa, T.; Uchida, K. Methylglyoxal Modification of Protein CHEMICAL AND IMMUNOCHEMICAL CHARACTERIZATION OF METHYLGLYOXAL-ARGININE ADDUCTS. *J. Biol. Chem.* **1999**, *274* (26), 18492–18502. <https://doi.org/10.1074/jbc.274.26.18492>.
- (15) Gillery, P. Nonenzymatic Post-Translational Modification Derived Products: New Biomarkers of Protein Aging. *Journal of Medical Biochemistry* **2011**, *30* (3), 201–206. <https://doi.org/10.2478/v10011-011-0021-7>.
- (16) Renard, B.-L.; Boucherle, B.; Maurin, B.; Molina, M.-C.; Norez, C.; Becq, F.; Décout, J.-L. An Expeditious Access to 5-Pyrimidinol Derivatives from Cyclic Methylglyoxal Diadducts, Formation of Argpyrimidines under Physiological Conditions and Discovery of New CFTR Inhibitors. *European Journal of Medicinal Chemistry* **2011**, *46* (5), 1935–1941. <https://doi.org/10.1016/j.ejmech.2011.02.037>.
- (17) Shipanova, I. N.; Glomb, M. A.; Nagaraj, R. H. Protein Modification by Methylglyoxal: Chemical Nature and Synthetic Mechanism of a Major Fluorescent Adduct. *Archives of Biochemistry and Biophysics* **1997**, *344* (1), 29–36. <https://doi.org/10.1006/abbi.1997.0195>.
- (18) Wang, T.; Kartika, R.; Spiegel, D. A. Exploring Post-Translational Arginine Modification Using Chemically Synthesized Methylglyoxal Hydroimidazolones. *J. Am. Chem. Soc.* **2012**, *134* (21), 8958–8967. <https://doi.org/10.1021/ja301994d>.
- (19) Chen, Y.; Ahmed, N.; Thornalley, P. J. Peptide Mapping of Human Hemoglobin Modified Minimally by Methylglyoxal in Vitro. *Annals of the New York Academy of Sciences* **2005**, *1043* (1), 905–905. <https://doi.org/10.1196/annals.1333.119>.
- (20) Gao, Y.; Wang, Y. Site-Selective Modifications of Arginine Residues in Human Hemoglobin Induced by Methylglyoxal. *Biochemistry* **2006**, *45* (51), 15654–15660. <https://doi.org/10.1021/bi061410o>.
- (21) Schmidt, R.; Böhme, D.; Singer, D.; Frolov, A. Specific Tandem Mass Spectrometric Detection of AGE-Modified Arginine Residues in Peptides. *J. Mass Spectrom.* **2015**, *50* (3), 613–624. <https://doi.org/10.1002/jms.3569>.
- (22) Johansen, M. B.; Kiemer, L.; Brunak, S. Analysis and Prediction of Mammalian Protein Glycation. *Glycobiology* **2006**, *16* (9), 844–853. <https://doi.org/10.1093/glycob/cwl009>.
- (23) Xu, Y.; Li, L.; Ding, J.; Wu, L.-Y.; Mai, G.; Zhou, F. Gly-PseAAC: Identifying Protein Lysine Glycation through Sequences. *Gene* **2017**, *602*, 1–7. <https://doi.org/10.1016/j.gene.2016.11.021>.
- (24) Thornalley, P. J. Dicarbonyl Intermediates in the Maillard Reaction. *Annals of the New York Academy of Sciences* **2005**, *1043* (1), 111–117. <https://doi.org/10.1196/annals.1333.014>.
- (25) Rabbani, N.; Thornalley, P. J. Dicarbonyl Stress in Cell and Tissue Dysfunction Contributing to Ageing and Disease. *Biochemical and Biophysical Research Communications* **2015**, *458* (2), 221–226. <https://doi.org/10.1016/j.bbrc.2015.01.140>.
- (26) Fritz, K. S.; Petersen, D. R. An Overview of the Chemistry and Biology of Reactive Aldehydes. *Free Radic Biol Med* **2013**, *59*, 85–91. <https://doi.org/10.1016/j.freeradbiomed.2012.06.025>.

Response to Reviewers

- (27) Zheng, Q.; Omans, N. D.; Leicher, R.; Osunsade, A.; Agustinus, A. S.; Finkin-Groner, E.; D'Ambrosio, H.; Liu, B.; Chandarlapaty, S.; Liu, S.; David, Y. Reversible Histone Glycation Is Associated with Disease-Related Changes in Chromatin Architecture. *Nature Communications* **2019**, *10* (1), 1289. <https://doi.org/10.1038/s41467-019-09192-z>.
- (28) Zheng, Q.; Osunsade, A.; David, Y. Protein Arginine Deiminase 4 Antagonizes Methylglyoxal-Induced Histone Glycation. *Nature Communications* **2020**, *11* (1), 3241. <https://doi.org/10.1038/s41467-020-17066-Y>.
- (29) Bollong, M. J.; Lee, G.; Coukos, J. S.; Yun, H.; Zambaldo, C.; Chang, J. W.; Chin, E. N.; Ahmad, I.; Chatterjee, A. K.; Lairson, L. L.; Schultz, P. G.; Moellering, R. E. A Metabolite-Derived Protein Modification Integrates Glycolysis with KEAP1–NRF2 Signalling. *Nature* **2018**, *562* (7728), 600. <https://doi.org/10.1038/s41586-018-0622-0>.
- (30) Milo, R.; Jorgensen, P.; Moran, U.; Weber, G.; Springer, M. BioNumbers—the Database of Key Numbers in Molecular and Cell Biology. *Nucleic Acids Res* **2010**, *38* (Database issue), D750–D753. <https://doi.org/10.1093/nar/gkp889>.
- (31) Wang, T.; Douglass, E. F.; Fitzgerald, K. J.; Spiegel, D. A. A “Turn-On” Fluorescent Sensor for Methylglyoxal. *J. Am. Chem. Soc.* **2013**, *135* (33), 12429–12433. <https://doi.org/10.1021/ja406077j>.
- (32) Shaheen, F.; Shmygol, A.; Rabbani, N.; Thornalley, P. J. A Fluorogenic Assay for Methylglyoxal. *Biochemical Society Transactions* **2014**, *42* (2), 548–555. <https://doi.org/10.1042/BST20140028>.
- (33) Park, M.-J.; Lee, S. H.; Moon, S.-J.; Lee, J.-A.; Lee, E.-J.; Kim, E.-K.; Park, J.-S.; Lee, J.; Min, J.-K.; Kim, S. J.; Park, S.-H.; Cho, M.-L. Overexpression of Soluble RAGE in Mesenchymal Stem Cells Enhances Their Immunoregulatory Potential for Cellular Therapy in Autoimmune Arthritis. *Scientific Reports* **2016**, *6* (1), 35933. <https://doi.org/10.1038/srep35933>.
- (34) Yu, W.; Liu, X.; Feng, L.; Yang, H.; Yu, W.; Feng, T.; Wang, S.; Wang, J.; Liu, N. Glycation of Paraoxonase 1 by High Glucose Instigates Endoplasmic Reticulum Stress to Induce Endothelial Dysfunction in Vivo. *Scientific Reports* **2017**, *7* (1), 45827. <https://doi.org/10.1038/srep45827>.
- (35) Vicente Miranda, H.; Gomes, M. A.; Branco-Santos, J.; Breda, C.; Lázaro, D. F.; Lopes, L. V.; Herrera, F.; Giorgini, F.; Outeiro, T. F. Glycation Potentiates Neurodegeneration in Models of Huntington’s Disease. *Scientific Reports* **2016**, *6* (1), 36798. <https://doi.org/10.1038/srep36798>.
- (36) Stratmann, B.; Engelbrecht, B.; Espelage, B. C.; Klusmeier, N.; Tiemann, J.; Gawlowski, T.; Mattern, Y.; Eisenacher, M.; Meyer, H. E.; Rabbani, N.; Thornalley, P. J.; Tschoepe, D.; Poschmann, G.; Stühler, K. Glyoxalase 1-Knockdown in Human Aortic Endothelial Cells – Effect on the Proteome and Endothelial Function Estimates. *Scientific Reports* **2016**, *6* (1), 37737. <https://doi.org/10.1038/srep37737>.

Reviewers' Comments:

Reviewer #1:

Remarks to the Author:

This is second time authors came back justifying their fundamentally flawed experiments. I would again strongly suggest:

These experiments must be repeated with high purity MG

The studies in HEK293T cells need to be performed with analysis of the wild type proteome. By over expressing a single protein at extremely high super physiological concentration and only analyzing this, the author has produced extreme positive bias to the results to observe and achieve the preferred outcomes.

Authors are incorrect stating that numerous previous studies did not uncover the underlying chemical features that govern preferential glycation. On the contrary, features positively influencing reactivity with MG where identified: (i) microscopic PKa1, (ii) surface exposure² (ref 3 in supplementary information)

Ref

1. Naila Rabbani, Amal Ashour and Paul J Thornalley Mass spectrometric determination of early and advanced glycation in biology, *Glycoconj J* (2016) 33:553–568 DOI 10.1007/s10719-016-9709-8

2. Ahmed, N., Dobler, D., Dean, M. & Thornalley, P. J. Peptide Mapping Identifies Hotspot Site of Modification in Human Serum Albumin by Methylglyoxal Involved in Ligand Binding and Esterase Activity. *J. Biol. Chem.* 280, 5724–5732 (2005). 35

Reviewer #2:

Remarks to the Author:

Overall, the reviewers' concerns have suitably been addressed in the revisions. The authors now provide an additional extended passage that clarifies the motivation and significance of the in vivo studies utilizing a GFP fusion.

The authors provide point-by-point responses to the concerns of Reviewer 1 and explain clearly the precedence for the use of commercially available MGO in a broad set of previous studies. The authors have conformed to accepted practices in this regard.

With regard to differences in results between the antibody-based approach used in this study and an alternative experimental protocol evaluating impaired trypsin digestion, the authors note in their response that "We had additional discussion about this in earlier drafts of the manuscript, but it was removed due to space constraints, as we felt it was not as essential to our conclusions as some of our other results."

Because this additional discussion addresses some important aspects related to reproducibility, the authors should consider adding it to the supplementary information.

Reviewer #3:

Remarks to the Author:

The response to largely the only experiment I have suggested is still confusing, unless I am significantly missing how exactly this experiment was performed.

The authors took a peptide which they reported to be a "hit" (peptide 1) and a "non-hit" (peptide 3) and fused this to GFP.

This was then expressed in cells.

The authors then evaluated the extent of glycation on these short unstructured peptides, which they know, in a test tube, are either prone to glycation (peptide 1) or resistant (for lack of a better term) to glycation (peptide 3).

Unless the peptides are somehow folded into GFP, these short peptides are likely to remain unstructured. Short, (likely) unstructured peptides, even if they are expressed in cells, do not provide any novel information on what is governing selective glycation on a folded protein, which provides physiological relevance for the study. Does this sequence exist on proteins? Are these proteins more prone to glycation?

Lastly, the repeated references to the previous works makes this manuscript seem like an incremental extension of the previous publication.

That being said, aside from the GFP experiment, the experiments were very well done. The figures are top-notch and the manuscript is very well-written. I would like to commend the authors, particularly the trainees, for enduring through this process.

Response to Reviewers

Reviewer #3

1. *"The response to largely the only experiment I have suggested is still confusing, unless I am significantly missing how exactly this experiment was performed. The authors took a peptide which they reported to be a "hit" (peptide 1) and a "non-hit" (peptide 3) and fused this to GFP. This was then expressed in cells.*

The authors then evaluated the extent of glycation on these short unstructured peptides, which they know, in a test tube, are either prone to glycation (peptide 1) or resistant (for lack of a better term) to glycation (peptide 3).

Unless the peptides are somehow folded into GFP, these short peptides are likely to remain unstructured. Short, (likely) unstructured peptides, even if they are expressed in cells, do not provide any novel information on what is governing selective glycation on a folded protein, which provides physiological relevance for the study."

We thank Reviewer 3 for further sharing their thinking regarding the GFP experiment that we performed. We completely agree that the fused sequences from peptide **1** and peptide **3** likely remain unstructured even when fused to GFP. However, we strongly disagree with the interpretation that this experiment *"do[es] not provide any novel information on what is governing selective glycation on a folded protein, which provides physiological relevance for the study."* To the contrary, in our view, this experiment provides a significant and novel result, clearly demonstrating that the sequence effects that control glycation *in vitro* are also able to control glycation for full-length proteins present within mammalian cells, an environment which contains countless competing factors (e.g., metabolites, native proteins, etc.) when compared to a highly controlled *in vitro* reaction. This was a purposeful choice in our experimental design, because had we decided to graft the peptide **1** and **3** sequences into a structured region of a particular protein, the confounding influence of that structure could have drastically influenced the glycation outcome in an (as of yet) unpredictable manner that would not be helpful for understanding if the sequence effects that we identified are physiologically relevant. In contrast, the experiment that we performed addresses only the most critical question: Are the trends seen *in vitro* relevant to glycation of intact proteins by MGO in mammalian cells? The experiment we performed directly addresses the question of whether or not the rules that control selective glycation of peptides *in vitro* also remain relevant for controlling the glycation of proteins expressed in cells. This approach therefore allowed us to draw the strongest conclusions about the specific *in vitro* results identified in this manuscript, while complementing and not retreading our previously published work.

We agree with Reviewer 3 that the remaining open questions about the contribution of structure warrant further investigation. Ongoing work in our lab seeks to determine how the trends observed *in vitro* are affected by secondary structure, tertiary structure, T_m , or other properties of folded proteins, and how they could be used to predict or control glycation on relevant cellular targets. Still, within the context of this study, the experiment we performed (Fig. 7 in the revised manuscript) does indeed demonstrate that our findings about the features that control selective glycation for peptides *in vitro* remain applicable on full-length intact proteins in living cells, and we respectfully suggest that we have already met the bar for demonstrating the physiological relevance of our findings.

We recognize that some of this confusion may stem from our original explanation of the experiment we performed. In particular, earlier versions of the manuscript used the word "folded" in the description of the GFP-**1** and GFP-**3** proteins, which may have inadvertently and incorrectly implied that we were attempting to evaluate the role of structure in influencing glycation outcomes. To avoid any confusion about and/or overstatement of the experiment we performed, we have removed all instances of the word "folded" from the description of the GFP experiment. In the revised manuscript, we have also explicitly described the "molecular features" or "trends" that control glycation as

Response to Reviewers

“sequence effects”, which serves to clarify both the appropriateness and the need for the experiment that we performed. Accordingly, the experiment we performed is a beautifully direct way to evaluate the transferability and applicability of our *in vitro* findings on peptides to proteins in a cellular context. ***In addition to the aforementioned revisions, we have also reorganized the discussion section and added significant new text that addresses the limitations of the experiment we performed and points to areas of further study.***

2. *“Does this sequence exist on proteins? Are these proteins more prone to glycation?”*

We share Reviewer 3’s interest in this question. We did indeed run the LESRHYA sequence through BLAST. However, we found that there are no human proteins that share 100% identity with it. There were a handful of proteins with roughly 85% identity to that sequence, some of which could perhaps be interesting to pursue. However, as past proteomics studies,¹⁻⁴ and our own prior work⁵ have clearly demonstrated that there is no ‘consensus sequence’ that controls selective glycation, we were concerned about pursuing this approach as it unavoidably conflates multiple questions about the differential effects of sequence and structure on glycation outcomes. Given the complex nature of glycation, it would simply not possible to address both sequence and structural effects as “add-on” experiments to the current manuscript. As the Reviewers note, answering these lingering questions about the influence of structure on the sequence effects we’ve identified will require extensive additional work, and we have now added these as explicit areas for future study in the *Discussion* section, as mentioned above. We are certainly eager to answer those questions, and this work is already underway in our lab, but it represents years of further work that is outside of the scope of the current manuscript. Thus, conclusively determining if the sequence effects seen *in vitro* are relevant to glycation of intact proteins by MGO in mammalian cells was the most important and relevant for this current work, and that is why the approach using GFP fusions is ideally suited for our current study.

3. *“Lastly, the repeated references to the previous works makes this manuscript seem like an incremental extension of the previous publication.”*

We thank Reviewer 3 for bringing this to our attention, as that was certainly not our intent. We had added discussion in our prior revision to provide much-needed context for the GFP experiment we chose to pursue. However, we may have overcorrected by providing too much detail about the findings of our prior work. ***To address this, we have removed the description of our prior work and streamlined the discussion of the results of our GFP experiment.*** We have also reorganized the *Discussion* section to better convey the ways in which this work represents a significant advance from our prior study.

4. *That being said, aside from the GFP experiment, the experiments were very well done. The figures are top-notch and the manuscript is very well-written. I would like to commend the authors, particularly the trainees, for enduring through this process.*

Thank you – these comments are greatly appreciated, especially by the trainees!

Reviewer #2

1. *“Overall, the reviewers’ concerns have suitably been addressed in the revisions. The authors now provide an additional extended passage that clarifies the motivation and significance of the *in vivo* studies utilizing a GFP fusion. The authors provide point-by-point responses to the concerns of Reviewer 1 and explain*

Response to Reviewers

clearly the precedence for the use of commercially available MGO in a broad set of previous studies. The authors have conformed to accepted practices in this regard."

We greatly appreciate these comments from Reviewer 2 and we are pleased that our prior revision has suitably addressed the lingering Reviewer comments from our prior submission.

2. *"With regard to differences in results between the antibody-based approach used in this study and an alternative experimental protocol evaluating impaired trypsin digestion, the authors note in their response that "We had additional discussion about this in earlier drafts of the manuscript, but it was removed due to space constraints, as we felt it was not as essential to our conclusions as some of our other results." Because this additional discussion addresses some important aspects related to reproducibility, the authors should consider adding it to the supplementary information"*

This is a great point. **We have added some discussion on this point to the figure caption in Supplementary Fig. 7.** Thank you for the suggestion!

Reviewer #1

1. *"This is second time authors came back justifying their fundamentally flawed experiments. I would again strongly suggest: These experiments must be repeated with high purity MG"*

As mentioned by Reviewer 2, we have conformed to accepted practices within the glycation field. A number of our studies were performed with purified AGE-modified peptides, and some were performed in the complex environment of a living mammalian cell. In both cases, our additional results supported our initial *in vitro* findings, eliminating any concerns about the source of MGO interfering with our results. Reviewer 1 has not provided a compelling or specific argument suggesting otherwise, despite raising this point on each revision. Numerous recent publications in this journal^{6,7} and other top journals^{8,9} have performed well-controlled studies using the same commercial source of MGO without further purification, and we have published our prior work in *Angewandte Chemie*⁵ using these same materials. Thus, the commercial source of MGO that we used for our *in vitro* studies is appropriate for generating useful information about how selective glycation is controlled, both *in vitro* and in a cellular environment.

2. *"The studies in HEK293T cells need to be performed with analysis of the wild type proteome. By over expressing a single protein at extremely high super physiological concentration and only analyzing this, the author has produced extreme positive bias to the results to observe and achieve the preferred outcomes."*

We strongly object to Reviewer 1's claim that our experiment was biased. Overexpression of proteins in mammalian cells is widely used across many disciplines, including cell biology and chemical biology. Still, it seems that Reviewer 1 is once again suggesting that overexpression of a protein in a mammalian cell an inappropriate experimental approach for reasons that remain unclear.

In this work, we compared the glycation of two different overexpressed GFP fusions (GFP-1 and GFP-3) to one another. We confirmed by western blot that GFP-1 and GFP-3 were expressed at comparable levels. Thus, **any differences in their glycation can and should be attributed directly to the differences in the GFP-1 and GFP-3 peptide sequences.** The design of the experiment is completely appropriate and unbiased.

3. *"Authors are incorrect stating that numerous previous studies did not uncover the underlying chemical features that govern preferential glycation. On the contrary, features positively influencing reactivity with MG where identified: (i) microscopic PKa1, (ii) surface exposure2 (ref 3 in supplementary information).
Ref*

1. Naila Rabbani, Amal Ashour and Paul J Thornalley Mass spectrometric determination of early and advanced glycation in biology, Glycoconj J (2016) 33:553–568 DOI 10.1007/s10719-016-9709-8

Response to Reviewers

2. Ahmed, N., Dobler, D., Dean, M. & Thornalley, P. J. *Peptide Mapping Identifies Hotspot Site of Modification in Human Serum Albumin by Methylglyoxal Involved in Ligand Binding and Esterase Activity. J. Biol. Chem. 280, 5724–5732 (2005).*”

We fully recognize that there have been many past efforts to identify the features that guide selective glycation on protein substrates, and this prior work has established that glycation occurs selectively for many proteins. However, those studies have rationalized findings individually without further experimental validation. For example, in the second article mentioned by Reviewer 1 (ref. 25 in our revised manuscript), Ahmed *et al.* reported preferential glycation of Arg410 in human serum albumin (HSA) by MGO. They computationally modeled the predicted surface exposure for each Arg in HSA and found that glycation occurred at some Arg that were largely solvent accessible, but also at some Arg that were less solvent exposed (including Arg410). They speculated that interactions between Arg and a nearby Asp “*may stabilize the guanidinium group and decrease its reactivity for surface exposed Arg that remained unmodified.*” While we agree this is completely plausible, this article does not provide any direct experimental evidence in support of their claims that “*surface exposure and neighboring group effects on the basicity of arginine residues may therefore account for the selectivity of glycation by methylglyoxal in HSA*”. Furthermore, based on an analogy to the HSA esterase mechanism, the authors proposed that MGO was activated by an initial hemiacetal with a nearby Tyr, thereby facilitating attachment of MGO to Arg410. However, this proposal was not validated experimentally. Taken together, it is our view that this prior study fell short of uncovering “*the underlying chemical features that govern preferential glycation.*”

Additionally, the first reference mentioned by Reviewer 1 is a review article citing many of the same articles that we have included in our manuscript, supplementary materials, and past responses to the reviewers. One prevailing theme summarized in that review article is that “*selectivity for sites of glycation is determined by the reactivity of the lysine, arginine or N-terminal residue under consideration. This is linked to: (i) microscopic pK_a of the residue being modified, (ii) surface exposure of the modification site...*” We agree that there have been many *ideas* put forth about the features that might promote glycation but few, if any, have been experimentally confirmed (as described above). Moreover, each study has been limited in scope and/or performed using conditions that are not directly comparable. Thus, these remain isolated reports that do not provide a unified understanding of the features that promote selective glycation. For instance, Rabbani *et al.* suggest that a lowered Arg pK_a has a positive benefit on glycation, though the experiments profiled in the review article report glycation at some Arg with lowered (predicted) pK_a and some Arg with elevated (predicted) pK_a. To us, the data presented do not appear to reconcile with the conclusions that were put forth.

While we agree that a lowered Arg pK_a could facilitate the first step of the glycation mechanism, our rigorous experiments have clarified that glycation—particularly MGH-1 glycation—is more likely driven by nearby polar and ionizable groups surrounding a reactive Arg, which are able to accelerate later, rate-determining reactions and rearrangements to stable AGEs. Indeed, Rabbani *et al.* also mention the possibility that “*a proximate conjugate base catalyzing the dehydration step involved in FL and MG-H1 residue formation*” may influence glycation selectivity. However, they cite a single study that monitored glycation by glucose (to form the Amadori product, fructosyl-lysine (FL)) for a set of 5 helical peptides,¹⁰ and we looked into the other references cited in Rabbani *et al.*, but most were review articles^{11,12} and we could not find a study (other than ours) that has provided experimental data in support of this for MGH-1 formation. It is therefore our opinion that Reviewer 1 is vastly overstating the findings of these prior works, and we respectfully suggest that our study provides the type of convincing, high-quality, rigorous experimental data that has been sorely lacking in prior publications in this research area.

Response to Reviewers

References:

- (1) Bilova, T.; Paudel, G.; Shilyaev, N.; Schmidt, R.; Brauch, D.; Tarakhovskaya, E.; Milrud, S.; Smolikova, G.; Tissier, A.; Vogt, T.; Sinz, A.; Brandt, W.; Birkemeyer, C.; Wessjohann, L. A.; Frolov, A. Global Proteomic Analysis of Advanced Glycation End Products in the Arabidopsis Proteome Provides Evidence for Age-Related Glycation Hot Spots. *J. Biol. Chem.* **2017**, *292* (38), 15758–15776. <https://doi.org/10.1074/jbc.M117.794537>.
- (2) Schmidt, R.; Böhme, D.; Singer, D.; Frolov, A. Specific Tandem Mass Spectrometric Detection of AGE-Modified Arginine Residues in Peptides. *J. Mass Spectrom.* **2015**, *50* (3), 613–624. <https://doi.org/10.1002/jms.3569>.
- (3) Johansen, M. B.; Kiemer, L.; Brunak, S. Analysis and Prediction of Mammalian Protein Glycation. *Glycobiology* **2006**, *16* (9), 844–853. <https://doi.org/10.1093/glycob/cwI009>.
- (4) Xu, Y.; Li, L.; Ding, J.; Wu, L.-Y.; Mai, G.; Zhou, F. Gly-PseAAC: Identifying Protein Lysine Glycation through Sequences. *Gene* **2017**, *602*, 1–7. <https://doi.org/10.1016/j.gene.2016.11.021>.
- (5) Sjoblom, N. M.; Kelsey, M. M. G.; Scheck, R. A. A Systematic Study of Selective Protein Glycation. *Angewandte Chemie International Edition* **2018**, *57* (49), 16077–16082. <https://doi.org/10.1002/anie.201810037>.
- (6) Zheng, Q.; Omans, N. D.; Leicher, R.; Osunsade, A.; Agustinus, A. S.; Finkin-Groner, E.; D'Ambrosio, H.; Liu, B.; Chandarlapaty, S.; Liu, S.; David, Y. Reversible Histone Glycation Is Associated with Disease-Related Changes in Chromatin Architecture. *Nature Communications* **2019**, *10* (1), 1289. <https://doi.org/10.1038/s41467-019-09192-z>.
- (7) Zheng, Q.; Osunsade, A.; David, Y. Protein Arginine Deiminase 4 Antagonizes Methylglyoxal-Induced Histone Glycation. *Nature Communications* **2020**, *11* (1), 3241. <https://doi.org/10.1038/s41467-020-17066-y>.
- (8) Maksimovic, I.; Zheng, Q.; Trujillo, M. N.; Galligan, J. J.; David, Y. An Azidoribose Probe to Track Ketoamine Adducts in Histone Ribose Glycation. *J. Am. Chem. Soc.* **2020**, *142* (22), 9999–10007. <https://doi.org/10.1021/jacs.0c01325>.
- (9) Lee, J. H.; Parveen, A.; Do, M. H.; Kang, M. C.; Yumnam, S.; Kim, S. Y. Molecular Mechanisms of Methylglyoxal-Induced Aortic Endothelial Dysfunction in Human Vascular Endothelial Cells. *Cell Death & Disease* **2020**, *11* (5), 1–15. <https://doi.org/10.1038/s41419-020-2602-1>.
- (10) Venkatraman, J.; Aggarwal, K.; Balaram, P. Helical Peptide Models for Protein Glycation: Proximity Effects in Catalysis of the Amadori Rearrangement. *Chemistry & Biology* **2001**, *8* (7), 611–625. [https://doi.org/10.1016/S1074-5521\(01\)00036-9](https://doi.org/10.1016/S1074-5521(01)00036-9).
- (11) Blom, N.; Sicheritz-Pontén, T.; Gupta, R.; Gammeltoft, S.; Brunak, S. Prediction of Post-Translational Glycosylation and Phosphorylation of Proteins from the Amino Acid Sequence. *Proteomics* **2004**, *4* (6), 1633–1649. <https://doi.org/10.1002/pmic.200300771>.
- (12) Rabbani, N.; Thornalley, P. J. Methylglyoxal, Glyoxalase 1 and the Dicarbonyl Proteome. *Amino Acids* **2012**, *42* (4), 1133–1142. <https://doi.org/10.1007/s00726-010-0783-0>.